# Inferring internal states across mice and monkeys using facial features

Alejandro Tlaie [1,2], Muad Y. Abd El Hay [1], Berkutay Mert[1], Robert Taylor [1], Pierre-Antoine Ferracci [1], Katharine Shapcott [1], Mina Glukhova[1], Jonathan W. Pillow [3], Martha N. Havenith [1,4] & Marieke L. Schölvinck[1,4] ✉

Animal behaviour is shaped to a large degree by internal cognitive states, but it is unknown whether these states are similar across species. To address this question, here we develop a virtual reality setup in which male mice and macaques engage in the same naturalistic visual foraging task. We exploit the richness of a wide range of facial features extracted from video recordings during the task, to train a Markov-Switching Linear Regression (MSLR). By doing so, we identify, on a single-trial basis, a set of internal states that reliably predicts when the animals are going to react to the presented stimuli. Even though the model is trained purely on reaction times, it can also predict task outcome, supporting the behavioural relevance of the inferred states. The relationship of the identified states to task performance is comparable between mice and monkeys. Furthermore, each state corresponds to a characteristic pattern of facial features that partially overlaps between species, highlighting the importance of facial expressions as manifestations of internal cognitive states across species.

In the wild, all mammals engage in similar foundational behaviours: they all hunt or forage for food, sleep, mate, avoid predators, and explore their environment, to name just a few. None of these behaviours can be simply explained as a passive reaction to environmental input; rather, they are crucially shaped by dynamic fluctuations in internal states such as satiety, alertness, curiosity or attention[1,2]. So, if fundamental behaviours are comparable across species, how similar are the internal states that drive them? Is 'attention' in a monkey the same as 'attention' in a mouse?

The common approach to investigate internal states has been a reductionist one: highly restrictive tasks featuring simplified stimuli and narrow behavioural repertoires (e.g. button presses), with little room for fluctuations over time[3–5]. What is more, experimental paradigms diverge widely depending on the species under study. For example, attention studies in primates typically require subjects to fixate on a central fixation point while paying attention to a peripheral stimulus that might subtly change its appearance[6,7]. Attention studies

in rodents, on the other hand, typically use the 5-choice serial reaction time task (5CSRTT), in which the subject is required to scan five apertures for the presentation of a brief light stimulus, and then navigate towards the light source[8,9]. Even though the behaviour associated with high attention, i.e. short reaction times and accurate responses, is the same in both cases, clearly these tasks are too different to draw any meaningful cross-species comparisons when it comes to the behavioural dynamics and neuronal mechanisms they may engage.

Breaking away from this restrictive regime towards studying internal states as they occur naturally is tricky. To tackle this challenge successfully, an ideal behavioural paradigm needs to (1) rely on innate, naturalistic behaviours to accurately reflect spontaneously occurring rather than training-induced internal states[10], (2) identify internal states in a data-driven way that is not restrained by (potentially anthropomorphising) concepts of cognitive processing imposed by the researcher, and (3) track the evolution of internal states over time

[1]Ernst Strüngmann Institute for Neuroscience in cooperation with the Max Planck Society, Frankfurt am Main, Germany. [2]Laboratory for Clinical Neuroscience, Universidad Politécnica de Madrid, Madrid, Spain. [3]Princeton Neuroscience Institute, Princeton University, Princeton, NJ, USA. [4]These authors contributed equally: Martha N. Havenith, Marieke L. Schölvinck. ✉e-mail: marieke.scholvinck@esi-frankfurt.de

to capture their intrinsically dynamic nature. For this, binary metrics of behaviour such as a button presses or nose pokes will not suffice; rather, precise, multi-parametric behavioural tracking is needed to generate time-resolved analyses that extract underlying cognitive states from the measured behavioural parameters moment by moment[11-13].

Recent technological advances have opened up new avenues to achieve these goals in a principled way. Virtual reality (VR) environments, for instance, allow researchers to create immersive yet highly controlled experimental settings that can be tailored to different species' intrinsic sensory capacities and behavioural repertoires[14,15]. Importantly, this maximizes adaptability across species, opening up the unique opportunity to record directly comparable behaviours in different species. At the same time, advances in deep-learning algorithms enable us to dynamically track ongoing changes in body movement and posture from video footage[16-18]. This allows for the ongoing and time-resolved tracking of behavioural dynamics—a fundamental prerequisite if we aim to identify the spontaneous emergence of internal cognitive and emotional states[19,20].

In this study, we leverage these technological breakthroughs to infer and directly compare the spontaneously occurring internal states of two species commonly studied in neuroscience - macaques and mice. Specifically, we combine a highly immersive and naturalistic VR foraging task solved well by both species[21-24], with a state-of-the-art deep learning tool that allows for precise, automated tracking of behavioural features. The features extracted in this way then serve as inputs to a Markov-Switching Linear Regression (MSLR) model[25], which infers time-varying internal states across trials.

Importantly, such single-trial inference of internal states is only meaningful if the behavioural markers it relies on are not indirectly tracking the concrete motor outputs required for task performance. For instance, task-related motor output such as preparatory paw movements might trivially predict a hit trial, and a lack of such movements might predict a miss trial. To ensure that the behavioural parameters we chose would truly reflect internal processing, we focused on the animals' facial expressions.

While facial expressions have long been thought to only play a role in highly visual and social species like monkeys and humans[26-29], recent work has highlighted that also less social, less visual species like mice exhibit meaningful facial expressions[19,30]. These expressions seem to reflect fundamental emotions like pleasure, pain, disgust and fear in a way that is not only consistent within one species, but also readily translatable across species[31,32]. This argues for an evolutionary convergent role of facial expressions in reflecting (and potentially communicating) emotions.

Outside the realm of emotional processing, spontaneously occurring behavioural states have so far mainly been tracked using individual facial features to identify isolated cognitive processes. For instance, pupil size has been related to changes in attention and arousal in rodents[33-35], non-human primates[36-40], as well as humans[41,42]. Similarly, eye movements in monkeys and humans[43-45] and whisker movements in mice[46] have been used to track attention and decision-making.

In addition to such individual facial features, movement intensity across the entire face (regardless of individual features) has been used to delineate, for instance, the contribution of instructed versus uninstructed movements to task performance[47,48], and to explain neural activity[49,50]. Orofacial motion is also a key component in explaining trial-to-trial variability in dorsal cortex activity in mice[51], especially when mice are disengaged from an instructed task[48]. More detailed analysis into task performance states based on such facial features revealed that optimal visual discrimination coincides with average pupil size and moderate levels of facial movement[52].

While these findings highlight promising ways to track internal states—both in the emotional and cognitive domain - using facial expressions, it stands to reason that facial expressions contain richer and more complex information than what can be extracted using either individual facial features or overall orofacial movement. By focusing on a multi-faceted representation of entire facial expressions, we aim to for the first time map out the spectrum of spontaneously occurring internal states in a data-driven way that is directly comparable across species. Our approach of using facial expressions to infer spontaneously occurring internal states from natural behaviour constitutes a drastic departure from the classical approach, which instead imposes internal states through restrictive behavioural tasks (e.g. cued attentional shifts). By tying the results of this approach back to known relationships between internal states and overt behaviour, such as shorter reaction times during focused attention, these spontaneous, agnostically inferred internal states can be tentatively related to known cognitive processes such as attention and motivation. Importantly, this puts us in the unique position to directly compare spontaneously occurring internal states across species by minimizing confounds introduced by species-specific tasks and training procedures.

## Results
### Experimental set-up
To track and compare spontaneously occurring internal states of mice and macaques during the performance of the same naturalistic visual discrimination task, the animals were placed inside a custom-made spherical dome (Fig. 1A, top). On the inside of the dome, we projected a virtual reality (VR) environment using a custom-made toolbox called DomeVR[53]. The monkeys navigated through the VR environment manually using a trackball; the mice ran on a spherical treadmill, the movements of which were translated into VR movements (for details, see 'Methods'—Experimental Setup).

Two monkeys and seven mice were used in this study, comprising 18 and 29 experimental sessions (20,459 and 12,714 trials), respectively. The animals engaged in a simple, foraging-based two-choice perceptual decision task, in which they had to approach a target stimulus while avoiding a distractor stimulus, both of which were represented by natural leaf shapes integrated in a meadow landscape (Fig. 1A, bottom; see 'Methods'—Experimental paradigm). Such tasks can be performed successfully by both monkeys[23] and mice[21,54]. Their performance on this task was quantified first in terms of trial outcomes: hit (target stimulus reached), wrong (distractor stimulus reached), and miss (neither stimulus reached); as well as in reaction time (RT). For this, we identified turning points in the animals' running trajectories through the VR to define the moment when an animal decisively oriented itself towards one of the two potential targets (Fig. 1C; for details, see 'Methods'—'Reaction Time'). As Supplementary Fig. S1 shows, success rate and reaction times were largely comparable across species, although mice showed less consistent performance than monkeys, in terms of running trajectories, reaction times, and correct target choices. We hypothesize that this is due to the lack of fine motor control of the mice on the trackball.

As the animals were performing the task, we recorded their faces. For macaques, this was done by analysing video footage from one camera positioned frontally on the monkey's face, as well as eye tracking output (see 'Methods'—'Behavioural tracking'). For mice, we analysed video footage from one camera positioned on the side of the face (Fig. 1B). From these videos, we extracted facial features such as eyebrow, nose and ear movement using DeepLabCut (Fig. 1C; see 'Methods'—Facial key point extraction). For monkeys, we selected 18 features; for mice, 9 features (see 'Methods'—'Facial features for the full list of facial features').

For each trial, facial features were averaged over a time window of 250 ms before the stimuli appeared in the VR environment. This time window was chosen to maximize the interpretability of the inferred hidden states: as there is no task-relevant information available yet, presumably all of the facial expressions that the animals make are due

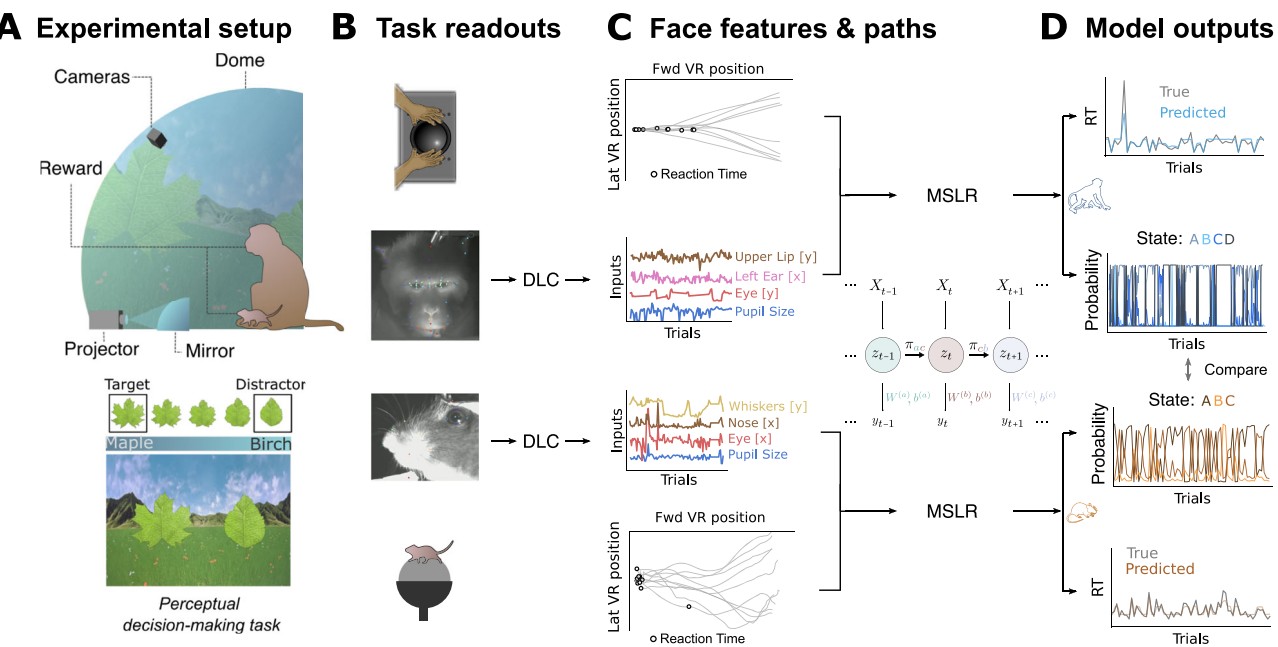

**Fig. 1 | Experimental setup and computational pipeline. A** Macaques and mice were seated inside a large dome on the inside of which a VR was projected via a curved mirror (top). They were rewarded for moving towards a spike-shaped leaf compared to a round-shaped leaf (bottom). **B** As the animals were engaged in the task, behavioural data were collected: movements of the trackball (top and bottom) and videos of their faces (middle). **C** Trackball movements were translated into paths through the virtual environment (top and bottom), from which reaction times were determined (see Methods). Individual facial features were automatically detected from the videos using DeepLabCut (DLC) and tracked over time (middle). **D** Facial features entered two separate Markov Switching Linear Regression (MSLR) models (one for each species), which yielded, for every trial, a predicted reaction time and internal state probabilities.

to internally generated processes, rather than being reflective of stimulus or task parameters.

## Model performance

The facial features extracted in this way were used as inputs to a Markov-Switching Linear Regression (MSLR) model (Fig. 1C; see Methods - Markov-Switching Linear Regression). We used a recently developed software package[55] to implement the MSLR. The MSLR manages to capture the non-stationarity and regime shifts often present in behavioural data[52,56–58], by flexibly accommodating complex temporal dynamics while keeping a relative simplicity, compared to deep learning-based methods[59]. Moreover, the MSLR is less data-hungry than other common data-driven models[60–62]. The MSLR uses the 'pre-stimulus' facial features in each trial to predict the animals' reaction time (RT) in the same trial by assuming 'hidden' states. Each hidden state implies a different linear relation between individual facial features and the subsequent RT in the same trial. For instance, in one hidden state, the RT might be best predicted by eyebrow movements, while in another, nose sniffing might be most predictive. We used cross-validation to select the number of states for each species (see below). For each trial, the model then outputs the predicted RT as well as the probability of each hidden state (Fig. 1D). The two models (one for mice, one for monkeys) were trained and tested on data from all individuals; Fig. S20 shows the outcomes of the models split by session and by individual and Fig. S8 shows the generalisation across mice.

Mathematically, this model takes the form:

$$RT_t = W_{z_t} \cdot x_t + \xi_{z_t},\qquad(1)$$

where $RT_t$ is the reaction time at trial $t$, $z_t$ is the state at trial $t$, $W_{z_t}$ are the regression weights for state $z_t$, $x_t$ is the vector of facial features at trial t, and $\xi_{z_t}$ is a zero-mean Gaussian noise with variance $\sigma_{z_t}$.

To test if this approach was appropriate for our behavioural recordings, we first checked if assuming the presence of multiple hidden states was in fact warranted by the data, or if they could also be

described by one constant, uniform relationship between facial expressions and RTs over time. To this end, we determined model performance when only one internal state was permitted (Fig. 2A; see also Fig. S14). For both species, the model's predictive performance was remarkably low under these circumstances - in fact, predictions were less accurate than random guessing.

Next, we quantified model performance for different numbers of hidden states—which is the main free parameter of the MSLR. Model performance was tested by using cross-validation (see Methods—Model tuning). Fig. 2A shows the cross-validated $R^2$ for different numbers of states and for optimal (solid line) and suboptimal (shaded area) constellations of hyperparameters. For both species, the cross-validated $R^2$ improved dramatically when allowing for more than one hidden state until reaching a plateau. Since the accuracy of RT predictions began to saturate with increased model complexity, we took the finite difference of the CV performance curve for each species and fixed the number of internal states at its maximum, in order to reach the optimal trade-off between predictive accuracy and model simplicity (Fig. 2A; see Fig. S11 for a more detailed explanation of this). This approach yielded a similar optimal number of hidden states for both species: for monkeys, the optimal number of states was 4, for mice it was 3. Tests on held-out data showed a similar performance (Fig. 2A, insets). What is more, model performance was consistently high across individual animals, and models trained in a leave-one-out analysis could also be successfully applied to data from animals they had not been trained on (see Supplementary Fig. S8). Together, these control analyses indicate that the model generalises robustly not just to held-out data but even across animals, suggesting that it captures shared behavioural features rather than individual idiosyncrasies. The CV procedure ended up selecting hyperparameters that mainly differed in the concentration parameter ($\alpha$), controlling how sparse each transition matrix was. For more details, see 'Methods'—"Model tuning".

In both species, and across all individual animals, our models yielded remarkably accurate trial-by-trial predictions of RT (see also Fig. S20), indicating that pre-trial facial expressions can indeed predict

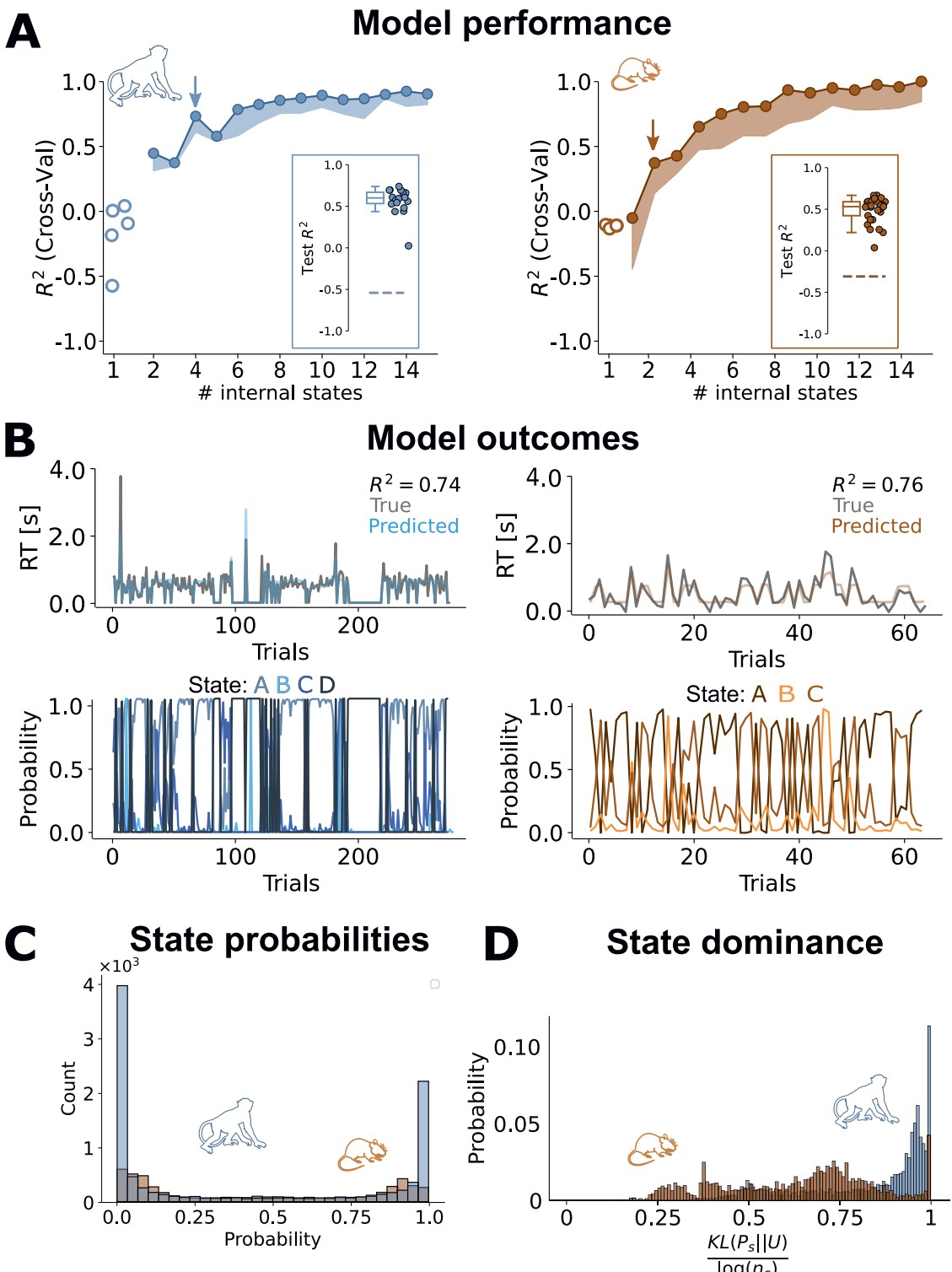

subsequent task performance (Fig. 2B, top row). It also suggests that the relation between facial features and task performance is dynamic rather than static over time, reflecting multiple underlying states. Finally, model performance was not impacted by removal of variables associated with pupil size, indicating that the performance did not rely trivially on known correlates of arousal (Fig. S16).

Is task performance dominated by a single state at any given moment, or do several states co-exist continuously? After fitting the model parameters, we used the model to identify the animal's internal state on a trial-by-trial basis. Note that the model does not allow for the

animal to be in multiple states at the same time; rather, it gives us probabilities telling how confident we can be about the state the animal is in on each trial. Specifically, we computed the posterior probability over states on each trial given all past and future observations. The probabilities of each state over time suggest that the model is highly confident about what state the animal is in on each trial (Fig. 2B, bottom row). These observations were confirmed by the highly bimodal distribution of these probabilities for both species (Fig. 2C). Crucially, in monkeys, this separation between high-certainty ($p_s \approx 1$) and low-certainty ($p_s \approx 1/n_s$) trials was particularly pronounced, while in

**Fig. 2 | Model performance and state probabilities. A** Cross-validation performance for various numbers of states ($n_s$), for macaques (left) and mice (right). Circles indicate the maximum CV $R^2$ and the shaded region extends until the 5th percentile over folds and repetitions ($n_{folds} = 5$, $n_{rep} = 5$) for each $n_s$. For both species, increasing the number of states improves model performance to a plateau at an $R^2 \approx 0.8$. *Lasso* is a regularized Linear Regression (i.e., a MSLR with 1 internal state). The arrows indicate the number of states we selected, based on the maximum difference of the CV performance curve (see Fig. S10). Insets show a box plot with the mean of the average model performance, and model performance for individual sessions as dots, for held out data at the selected number of states; dashed horizontal lines indicate the $99^{th}$ percentile of the surrogate performances (see Methods). Note that the shuffled $R^2$ is negative because only uncorrelated predictors are expected to be centred at 0, and due to finite sampling effects, there is always a non-zero correlation between the shuffling and the ground-truth. Furthermore, as we are dealing with skewed distributions (see Fig. S1), the null tendency is not captured by the mean, as assumed by the default $R^2$. **B** Predicted RTs (top) and state probabilities (bottom) for an example stretch of data (left, macaques; right; mice). **C** Probabilities of all states over all trials, regardless of state identity (blue, macaques; orange, mice). The bimodal distribution suggests that states are either absent or dominant on any given trial. **D** Kullback-Leibler divergence (KL) for monkey (blue) and mouse (orange) internal states. KL quantifies the difference between the posterior state probability under the model and the uniform distribution, normalizing by the number of states. A KL value close to 1 indicates maximally dissimilar distributions (i.e., only one present state at a time), while a value close to 0 indicates indistinguishable distributions (i.e., equally likely states).

mice, state probabilities were somewhat more mixed. Quantifying the single-trial certainty as measuring its difference with the uniform distribution –through the Kullback-Leibler divergence (KL)–corroborated these findings (Fig. 2D; Mann-Whitney U-test: $p = 1.11 \cdot 10^{-274}$). As such, the hidden states identified by our model seem to reflect largely mutually exclusive behavioural modes that animals switch in and out of. Given how consistently trials were dominated by one state, we chose to binarize hidden state outcomes by assigning each trial to its most probable hidden state.

### State dynamics
To explore if the hidden states showed attributes that could be reflective of internal cognitive states, we first characterized their temporal dynamics. To this end, we examined the frequency of state transitions in both species. The state transition matrices, which show how likely a trial of a given hidden state is followed by a trial of any (other or same) state (see Methods - Markov-Switching Linear Regression), revealed high values along the diagonal for macaques, indicating stable states that switched rather rarely. In mice, the diagonal of the transition matrix was slightly less pronounced, suggesting that hidden states in mice were less stable and more prone to transition than in macaques (Fig. 3A). This was confirmed by the learned hyperparameters of both models: These were highly comparable across mice and monkeys with the exception of the concentration parameter, which governs the sparseness of transition probabilities and was markedly lower for monkeys than mice (see Methods - Model tuning for details and for the values of the parameters we selected).

As a complementary analysis, we computed the dwell time for each state. This quantity is defined as the number of consecutive trials that a given state is occupied for before transitioning to a different state. As dwell times are dependent on absolute numbers of trials spent in each state (a state with very few trials overall cannot have long dwell times), we plotted the dwell times alongside the number of trials spent in each state (Fig. 3B). Supporting the previous observations, hidden states lasted generally longer in macaques than in mice (Mann-Whitney U-test; $n_{mac} = 4092$, $n_{mice} = 2543$ trials, $p = 0.0014$), suggesting that internal processing may be more steady in macaques. This is consistent with previous findings that behavioural dynamics may fluctuate faster in mice[34,63] than monkeys[64]. Apart from a genuine species-driven difference, this observation may also reflect the fact that monkeys are trained more extensively and may therefore have developed more stereotyped behavioural strategies than mice, which were trained more briefly.

### Hidden states as performance states
To link the identified hidden states more concretely to internal cognitive processing, we set out to investigate how each hidden state related to behavioural outcomes, starting with the RTs that the model was trained to predict. There are two potential scenarios for how the model might partition RT variability: on the one hand, it is possible that each hidden state covers the full range of RTs, but predicts them from a different constellation of facial features. Alternatively, each hidden state might 'specialize' on predicting specific ranges of RTs. For example, one hidden state might cover facial features that distinguish between fast and extremely fast RTs, while another state mainly predicts variations between slower RTs. This second scenario would make it more likely that the identified hidden states reflect genuinely distinct performance states.

To distinguish between these scenarios, we plotted the overall state-specific RT distributions, pooling trials across all sessions and animals, for each hidden state (Fig. 4A; top panel; Fig. S21 shows the same plot for individual sessions and animals). The resulting distributions support the second scenario: while one hidden state (state B in both monkeys and mice) covered a rather broad range of RTs, all other states showed a distinct profile of response speeds. This implies that the hidden states relate to distinct performance regimes (in this case, in terms of response speed), making them viable candidates for defining specific internal states of cognitive task processing.

To further probe the possible link of our internal states to known cognitive processes, we related all hidden states to the three possible trial outcomes of the task (hit, wrong, and miss; see Methods–Experimental Paradigm). Crucially, given that we trained the model to predict RTs, it never received any explicit information about trial outcome. Furthermore, RTs were only marginally related to trial outcomes (Fig. S1), so that trials with a specific RT would not be significantly more likely to result e.g. in a hit or a miss trial. Finally, as we only used information about facial features in the pre-stimulus phase of the trial to train the model, it cannot reflect stimulus features.

Even though information about trial outcomes was not part of the MSLR model, the resulting hidden states were consistently predictive of specific trial outcomes (Fig. 4A, bottom panel). For instance, in monkeys, trials that were classified as belonging to state C were most likely to result in a hit, while trials from state A often resulted in incorrect responses, even though the RT distributions of both states overlapped strongly. The same dynamic can be observed in states A and C in mice.

Adding more states to the models and subsequently predicting RTs and task outcome invariably resulted in several states strongly overlapping in their predictions. For example, using 10 states in monkeys and 8 states in mice (models with best cross-validated performance, see Fig. 2A) resulted in two groups (of 2 and 6 states) in monkeys and two groups (of 2 and 3 states) in mice, where the relationship of the states to RTs and task outcome was very similar (Figs. S10, S12). We therefore conclude that the task performance space is best covered by the three states for mice and four states for monkeys.

Linking internal states to RTs and trial outcomes revealed that individual states covered unique combinations of speed and accuracy. To visualize these combinations, we plotted mean RT per hidden state against the difference in probability of a hit versus a wrong trial for each state. Interestingly, the constellation of states in this space was comparable across species (Fig. 4B). Both mouse and monkey data

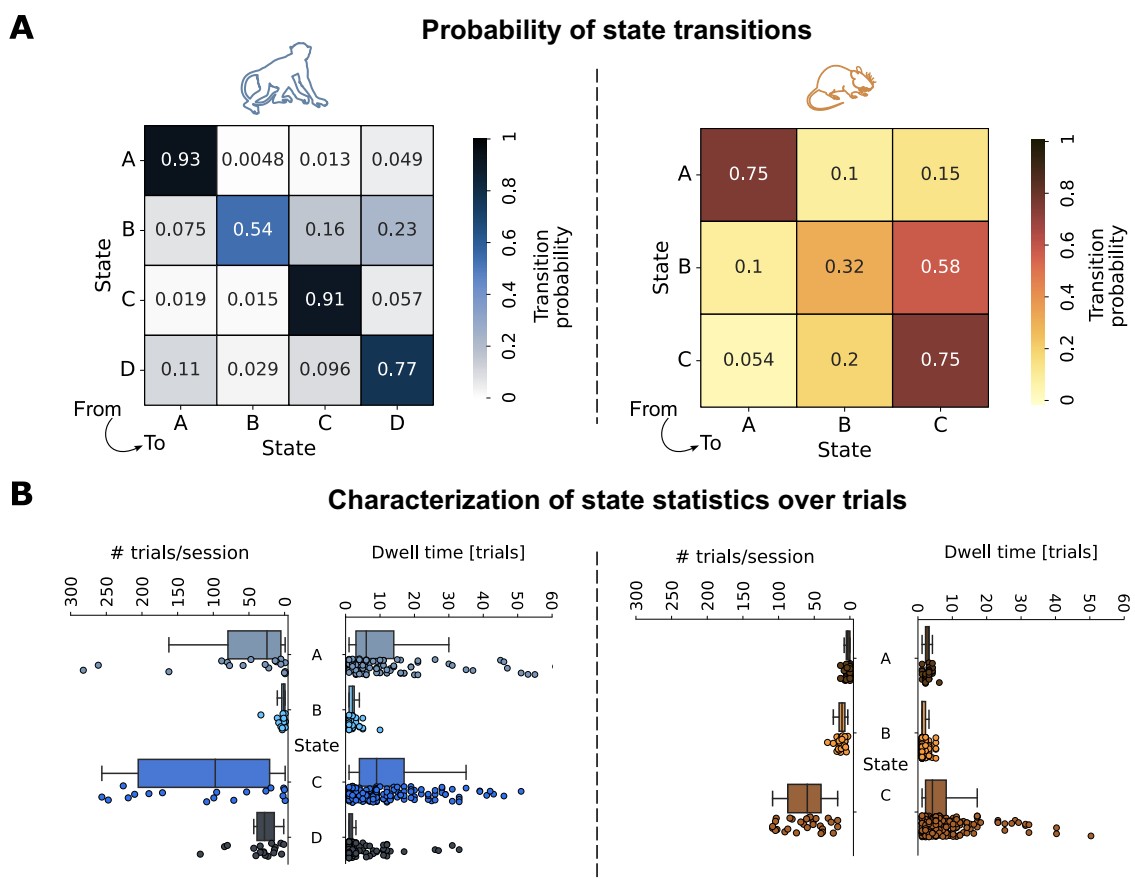

**Fig. 3 | State dynamics. A** State transition matrices for macaques (left) and mice (right), that show the probability, at any one trial, of transitioning from a certain state (rows) to any other state (columns). Transitions between different states (off-diagonal terms) are more frequent for mice than for macaques. **B** Macaques (left plots) spend more time than mice (right plots) in the same state, as measured by the dwell time (number of consecutive trials of each state being the most likely one;

$n_{mac} \in [90, 1930]$, $n_{mou} \in [110, 1160]$). Context for these dwell times is given by the trial counts (per session) on the left of each dwell time plot ($n_{mac} = 18$, $n_{mou} = 29$). Box plots reflect the median and 25th, 75th percentiles of the dwell time and number of trials per session, with whiskers showing $10^{th}$, $90^{th}$ percentiles; individual dots below the box plots reflect sequences of consecutive trials of a particular state and trial counts per session, respectively.

seem to generate one hidden state (state A in mice, state C in monkeys) that is associated with fast RTs and largely successful trial outcomes - a performance regime that could be interpreted as globally attentive. Conversely, state C and A in mice and monkeys, respectively, map onto rather fast yet often incorrect responses, potentially reflecting more impulsive decision-making[21,65]. Finally, state B for both species features particularly slow RTs, large RT variability, and mostly incorrect trials for mice and equally likely trial outcomes for monkeys, potentially signifying a state of global inattention[66,67]. The only state that appears in monkeys but not mice (state D) features no reactions at all (i.e. no change in path direction) and only misses; a sign of complete task disengagement.

**Relationship to facial features**

A final clue towards the interpretation of our internal states might be given by the facial features from which they are inferred. To explore this possibility, we plotted the regression weights of all facial features for the hidden states associated with hit, wrong, and miss trials (Fig. 5A; for the facial features comprising the fourth state in the monkey, see Fig. S13). These plots reveal highly distinct contributions of different facial features to each internal state. For example, in mice, pupil size and whisker movements predict reaction speed in the 'hit' state, whereas nose movements predict reaction speed in wrong and miss states. Similarly, in monkeys, large pupil size predicts fast reactions in hit and wrong states, but slow reactions in miss states, and eye movements play a strong predictive role in hit and wrong, but not in

miss states. One interpretation of these observations is that different sensory modalities may be more dominant in driving decision making (and thereby decision speed) in different states. Especially in mice, one of the hallmarks of the hit state is that it is the only state in which pupil size plays a decisive role, suggesting that in other states, behaviour may be less strongly driven by sensory sampling from the visual domain (and more by sampling e.g. from the olfactory domain).

Interestingly, the facial constellations predicting RTs in the states most closely associated with hit and wrong trials, respectively, are quite similar in monkeys, but not in mice. This may suggest that in monkeys, the behavioural state underlying correct and wrong trial outcomes may be a generally engaged and high-attention state, and hit or wrong outcomes are mainly dictated by visual difficulty than different internal cognitive state. In contrast, in mice, hit and wrong trials may be the product of more distinct underlying cognitive states, e.g. in terms of attentive capacity (see Supplementary Fig. S19 for a summarized visualization).

This notion becomes even more apparent when focusing only on the facial features that both species have in common (pupil size, and eye and nose movements), as shown in Fig. 5B. Surprisingly, the direct comparison of shared facial features across species shows that the contribution of individual facial features to the three performance states is highly overlapping, particularly in hit and miss states (Fig. 5B). Shuffling the features and computing the mouse-monkey overlap 10.000 times, revealed that this overlap is extremely strong in the hit state, quite strong in the miss state, and indistinguishable from chance

## A    Reaction Time distribution per state

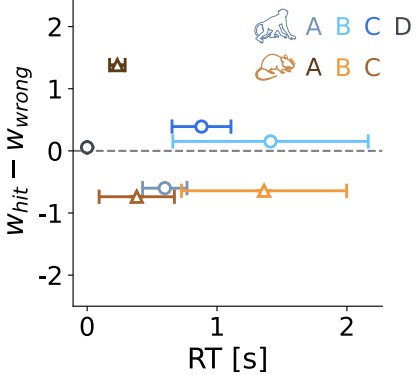

## B  State similarity between macaques and mice

**Fig. 4 | Internal states and task performance. A** (Top) Splitting the RTs over internal states shows large diversity for both macaques (left) and mice (right), from fast reaction-states to extremely slow ones. Box plots reflect the median RT and the 25th, 75th percentiles, with whiskers showing 10th, 90th percentiles; individual dots reflect trials ($n_{mac} \in [90, 1930]$, $n_{mou} \in [110, 1160]$). (Bottom) Correlations of state probabilities with the three task outcomes (hit, wrong, miss), for macaques (left) and mice (right). Black boxes indicate the states most strongly associated with a certain task outcome. **B** Conjunction of RT and excess likelihood of a hit outcome, for all states (blue circles, macaque; orange triangles, mouse) ($n_{mac} \in [90, 1930]$, $n_{mou} \in [110, 1160]$).

in the wrong state (at 97.9, 86.2, and 56.7 percentile of the distribution, respectively). This result further underlines the notion that, unlike hit- and-miss trials, wrong trials may be the outcome of genuinely distinct cognitive states in mice and monkeys. More globally, the strong match of facial features, especially in hit and miss states, indicates that facial expressions may be evolutionarily preserved across species not only in terms of emotional[32], but also in terms of cognitive and attentional processes.

Do the internal states reflect global facial constellations, or could they be inferred equally well from individual facial features such as pupil size? To answer this question, we plotted the contribution of individual facial features across all inferred internal states. (Fig. 5C shows the distribution of model weights across all facial features. Some features (e.g. the upper lip in monkeys or vertical whisker movement in mice) did not contribute significantly to predicted

performance in any internal state. For more prominently predictive facial features (e.g. right ear or pupil size in monkeys and vertical eye movements or nose movements in mice), there was a wide spread of model weights across states. This indicates that no single facial feature was predictive of performance across all states. Rather, in both species, reaction times are best predicted by a complex and variable con- stellation of facial features that cannot simply be reduced to individual components. This finding is further supported by the observation that different states are consistently distinguished by more than one facial feature (Fig. 5A; see also Fig. S9), and that removing one of the most prominent predictive facial features—pupil size—together with all facial variables significantly correlated to it, did not impair model performance in any way (see Figs. S15, S16).

Together, these results suggest that 1) holistic analysis of complex facial expressions is much more informative than analysis of one iso- lated facial feature such as pupil size and 2) the relationship between facial features and cognitive processing is not linear, but changes depending on the internal state that the animal is in. For instance, in a high-performance state, large pupil size may indeed predict trial suc- cess (as shown e.g. by refs. 33,36), whereas it may be irrelevant or anti- correlated in a low-performance state (see e.g. contribution of pupil size to hit versus miss states in monkeys, shown in Fig. 5A).

Lastly, we aimed to establish whether the constellations of facial features associated with each internal state were sufficiently distinct to infer internal states in individual trials. To this end, we classified the facial read-out from each trial as part of a hit, miss or wrong state, based on its correlation to the corresponding hit, miss, and wrong facial profiles (Fig. 5D). In both species, trials were classified as belonging to their rightful internal state far above chance level (Fig. 5D). This suggests that in both species, there are indeed char- acteristic facial expressions reflecting e.g. attentive or disengaged cognitive states, that are distinctive enough to be successfully identi- fied on a single-trial basis in a majority of trials.

### Influence of trial history

One reason why hidden states can predict trial outcomes so accurately despite not being trained on them in any way might be that pre-trial facial features are mostly a trivial consequence of the animal's trial history. For example, facial features might mainly reflect an animal still drinking reward from the previous trial, which might in turn raise motivation to perform correctly in the upcoming trial. In this case, facial features would merely be a particularly convoluted way of quantifying the previous trial outcome, and using it to predict upcoming performance, as has been achieved previously[68,69]. To account for this possibility, we trained an Auto-Regressive Hidden Markov Model (ARHMM) based on RTs (see Methods - ARHMM for details). As can be seen in Supplementary Fig. S6, the facial features model outperforms the ARHMM for all states, for both species.

As an extra control, we correlated each facial feature with the history of prominent task parameters, specifically two related to the directly previous trial (its outcome, which might affect motivation; and the location of its target, which might predict side biases), and two related to the overall session history (the cumulative amount of reward and the time that passed since the start of the session, as proxies for satiety and fatigue, respectively). Correlations between task variables and facial features were sparse in both species (Supplementary Fig. S17). In fact, attributes of the previous trial did not relate sig- nificantly to facial features at all, and more sustained session attributes modulated facial features merely somewhat. This suggests that facial features may be modulated by ubiquitous internal processes like fati- gue and satiety, which are in turn impacted by task parameters, but they are not a trivial reflection of task history. Rather, the fact that facial expressions are modulated by the overall task context makes them a more plausible reflection of realistic fluctuations in cognitive processing.

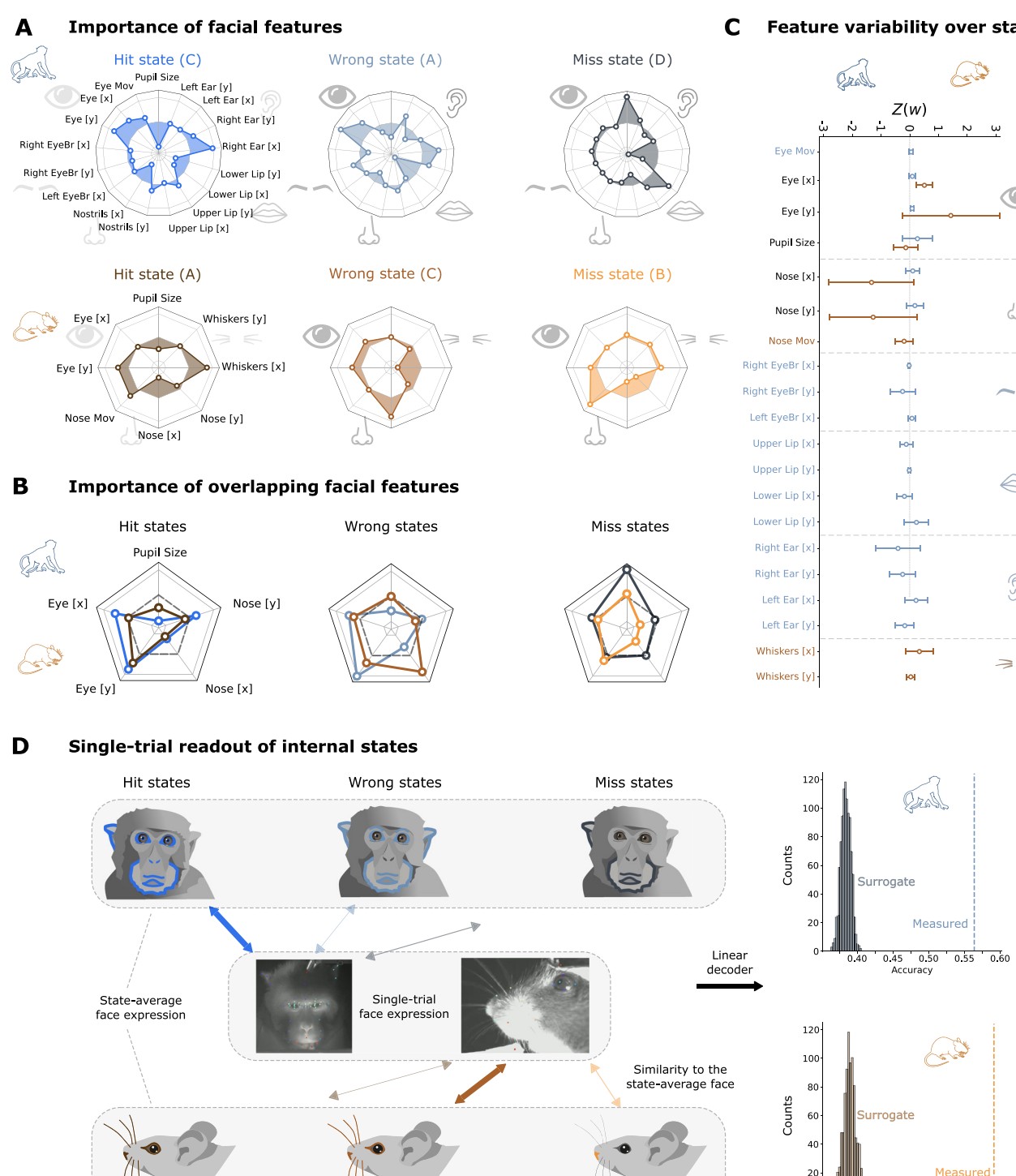

**Fig. 5 | Informativeness of facial features. A** Predictor weight of the facial features for the macaque (top) and mouse (bottom) model in the hit, wrong and miss states (see black boxes in Fig. 4A, bottom). Central circle indicates a predictor weight of zero; inside this circle are negative predictor weights, outside are positive weights. Note that a positive weight implies a negative relationship with performance and vice versa (for example, the strong negative weight on pupil size for monkeys in the hit state means that the bigger the pupil, the slower reaction times it predicts). Each state has its own characteristic facial expression pattern. **B** Plotting only the facial features that overlap between the species reveals strong similarities in their importance to the three performance states. **C** Variability of all facial features ($n_{mac}$ = 4, $n_{mou}$ = 3) over states. Although some features contribute more than others, clearly, all features contribute to the model distinguishing between the various internal states. We show the mean value ± SEM. **D** As seen in (**A**), each state has a distinctive facial expression pattern. Further supporting this, we show that we can decode states on single-trial states based on similarity to each state-averaged template.

## Discussion

Internal cognitive states are known to substantially shape overall brain activity[70,71] as well as behavioural decision making[1,2], yet they are notoriously difficult to identify. As a result, it is even less clear to what extent they converge across species. To infer hidden cognitive states in mice and monkeys, we harnessed an MSLR model[25] trained on their facial features while they were engaged in an immersive VR foraging task. Specifically, we trained the MSLR to predict an animal's reaction time (RT) in a given trial based on its facial expressions prior to stimulus presentation. For both species, RTs could be predicted on a trial by trial basis with high accuracy from preceding facial features only. This suggests that facial expressions reflect internal parameters that are directly relevant to task performance from moment to moment. These parameters were only minimally shaped by task history, suggesting that they were not a trivial reflection e.g. of the previous trial outcome.

Even more surprisingly, this approach revealed multiple distinct hidden states, which were characterized by equally distinct relationships between a complex constellation of facial features and subsequent task performance. In different states, performance seemed to be dominated by specific sensory modalities, e.g. eyes versus nose for hit versus wrong states in mice. This suggests that, depending on an animal's internal state, the relation between facial features and subsequent task performance can shift dramatically, and there are no individual facial features that reliably predict behaviour across different internal states. As such, our results emphasize how data-driven characterization of internal states can highlight crucial features that might be missed when using experimenter-defined states (see also ref. 72).

These findings stand in marked contrast to previous research studying mostly linear relationships between single facial features (e.g. pupil size or eye movements) and isolated cognitive states (e.g. attention) pre-defined by the experimenter[33,34,36–39,43–45]. They also go beyond recent studies highlighting the relationship between motion of the entire face (regardless of the facial features), and task performance (Musall et al. 2019; Hulsey et al., 2024) or neural activity (Syeda et al. 2023; Salkoff et al. 2020; Talluri et al. 2023). This highlights the fact that information reflected in the entire face is not only richer but also qualitatively different from information extracted from individual facial features.

Most importantly, the internal states revealed by the MSLR model mapped distinctly onto behavioural trial outcomes (i.e. hit, wrong and miss trials)—even though this information had been in no way part of the inputs the model received. This suggests that the hidden states highlighted by our model were not simply 'computational devices' increasing its predictive power. Instead, they appear to reflect genuine, dynamically fluctuating cognitive states, which result in distinctive behavioural outcome profiles.

Our findings were not dependent on the use of one specific MSLR model, as we repeated our analyses with a GLM-HMM model (training an individual model per animal and experimental session), with very similar results[73]. In two separate studies, such a GLM-HMM model was applied to behavioural choices in sensory discrimination tasks in mice. Both studies revealed distinct cognitive states that were identified as states of high or low engagement in the task[48,52]. The states were correlated with classic measures of arousal: pupil diameter, spontaneous facial movements and locomotion[52]. Moreover, brain wide imaging signals recorded during a disengaged state showed higher trial-to-trial variability, which was due to task-independent (partly facial) movements[48]. By taking into account comprehensive measurements of the entire face rather than mainly behavioural outcomes, our study could not only infer performance states on a single trial level (Fig. 5D), but also show that disengaged states can reflect the dominance of less relevant sensory modalities (Fig. 5A), and that similar behavioural states engage similar facial features in monkeys and mice (Fig. 5C),

suggesting strong evolutionary preservation of the underlying mechanisms.

Interestingly, despite the fact that the optimal number of states was determined separately for each species and in a purely data-driven way, our approach converged onto a low and noticeably similar number of internal states for both species: three states for mice (in agreement with[48,52]), four for macaques. How comparable are these internal states of mice and monkeys?

We found that in terms of the dynamics by which animals traversed different internal states, results diverged across species. Specifically, mice appeared to transition more frequently between states than monkeys. A control analysis that matched the number of subjects, trials and facial parameters across species before fitting the MSLR models showed that this difference is not a trivial result of divergences in data structure (see Supplementary Fig. S18). Given that mice have previously been shown to alternate between strategies during perceptual decision-making[56], this finding may point at a genuine difference in the cognitive dynamics of mice and monkeys. Alternatively, the prolonged training time of the monkeys compared to the mice may have given them the chance to converge on more stable behavioural strategies over the course of training[74,75]. Recordings from mice that experienced a more prolonged training scheme and/or from more naive monkeys will give fascinating insights into the role of expertise in fostering more stable transitions between cognitive states.

Beyond state dynamics, the constellation of behavioural profiles covered by different states was largely comparable across species[76,77]. Each hidden state predicted only a narrow range of reaction times; and when relating the inferred hidden states to task performance beyond the RTs that the model was trained to predict, we found that states mapped onto the behavioural outcomes (hit, wrong, miss) with distinct probabilities. Moreover, each hidden state covered unique combinations of RT ranges and trial outcomes (hit, miss and wrong trials), despite the fact that trial outcomes had not been part of the MSLR in any way. Specifically, both monkeys and mice display a state where trial outcome is typically slow and unsuccessful (which could be interpreted as 'inattentive'), as well as several states where performance is largely fast and correct, with a preference for thoroughness in one state, and a preference for speed (and potentially impulsivity) in the other. These states potentially map onto various levels of task-related attention, and further support the notion that classical concepts of attention can indeed reflect much of the internal structure of goal-directed behaviour, also in naturalistic settings.

Different internal states were associated with distinct constellations of facial features, as evidenced by the facts that states were highly separable based on the associated facial features, and that they could be successfully inferred through facial template matching on a single-trial level. This strong relation points to a role of facial expressions beyond emotional expression. Facial expressions have so far been mostly studied in a social or emotional context, and mostly in social species such as monkeys[29,78] and humans[26,79]. In mice, until recently facial expressions were thought to mainly reflect pain[30,80,81], until careful analyses using machine-learning algorithms identified their facial expressions as innate and accurate reflections of several emotional states as well[19,31,82]. Our results suggest that similarly to humans, facial expressions in monkeys and mice also convey cognitive and motivational variables such as focus or cognitive strain, even in the absence of a particular emotional or social context. What's more, the facial features predictive of comparable behavioural outcomes in mice and monkeys were highly overlapping (Fig. 5C). This suggests that the mechanisms mapping behavioural states onto facial expressions are robust and highly preserved across the evolutionary ladder.

The fact that such performance-related states are equally apparent in both species is intriguing, since one might have assumed that the

prominent differences between the two species (such as the acuity and dominance of their visual system) would imply that they are likely to solve tasks using different strategies. And while both species have been shown to experience spontaneously shifting internal states, such as slow fluctuations in attention[21,64,65,70], it has been difficult to make accurate comparisons between species based on available work. Studies of internal states have typically employed species-specific tasks that require extensive training[56,64], which may have imposed cognitive dynamics that would not occur in the wild. As such, similarities and differences in the internal states identified in different species might have reflected similarities and differences in behavioural settings and training schemes just as much as innate differences in cognitive processing. Our study minimizes such confounds compared to previous research[56,64], and thus adds one of the first rigorous estimates of which aspects of internal states truly generalize across species—and which do not. The fact that with this approach we were able to identify strong similarities in spontaneously occurring internal states across species opens the door to further inquiries into the common and evolutionary preserved principles of these states. Specifically, the spontaneous state fluctuations identified here can be important indicators of underlying brain-wide activity fluctuations that may well generalize across species to a large extent. Global internal states such as arousal, motivation, and attention typically manifest themselves via brain-wide dynamics, and it will be an exciting endeavour to investigate how well those neuronally defined internal states correspond to the ones we here identified behaviourally.

The MSLR model that we used yielded single estimates of the internal states per trial, and our subsequent analysis using 'facial templates' of internal states for single-trial predictions of behavioural outcomes was highly successful in both species (Fig. 4). This constitutes a great basis for accurate, time-resolved tracking of ongoing dynamics in internal states, which can be further extended in future, using MSLR models with higher temporal resolution. Such MLSR models will be able to align identified internal states with specific events within each trial, such as the appearance and disappearance of stimuli, thereby allowing for more precise characterisation of their dynamics and functional roles.

Perhaps most importantly, such a time-resolved MSLR would also allow us to link cognitive processes to neural activity on a moment-by-moment basis, without the need for repeatedly presenting identical trials and then doing extensive post-hoc averaging. As the MSLR model yields a time-resolved estimate of cognitive states, these time courses can be directly compared to continuous neural activity. As such, this approach opens up a much more naturalistic view of the neuro-behavioural dynamics involved in spontaneous cognitive states than traditional approaches can offer[47,57].

These findings suggest that in an ecologically valid framework that applies across species (in this case, a foraging-based task set in a naturalistic, immersive visual environment), many features of cognitive processing are more similar than classical paradigms might have suggested. At the same time, presumably genuine cross-species differences, e.g. in the transition frequency between cognitive states, also become more apparent.

In summary, we have shown here that in both monkeys and mice, facial features can be used to infer internal cognitive states and to track their spontaneous dynamics over time. With this approach, we find that the basic attributes of such internal states map onto known cognitive states such as attention in both species in a translatable way, but that the dynamics by which mice and monkeys traverse these states are somewhat different. This highlights the crucial importance of using naturalistic behavioural paradigms, especially in cross-species research, in order to discern truly species-specific results from differences induced by restrictive testing methods.

## Methods

### Animals

This study includes data from two male macaques (*Macaca mulatta*) and six male Black6 mice (*Mus musculus*). The monkeys were housed together in a spacious outside and inside area, connected via a small flap door and both containing environmental enrichment. Mice were kept in day-night reversed housing and were initially group-housed, but housed individually once behavioural training commenced to ensure correct food restriction. Single-housed animals were given 'playtime' in a large play cage shared with litter mates following their behavioural training in order to counteract stressful effects of single housing. Mouse cages were placed in a room with a temperature of 21.5 °C and 55 percent humidity, and contained environmental enrichment such as running wheels. All procedures were approved by the regional authorities (*Regierungspräsidium Darmstadt*) under the authorization number F149/2000 and were performed in accordance with the German law for the protection of animals and the 'European Union's Directive 2010/63/EU'.

### Surgical procedures

All animals were fitted with custom-milled headposts for the purpose of head fixation during this experiment. The headpost design and implant procedures for the macaques have been extensively discussed in ref. 83. Briefly, a four-legged titanium baseplate was screwed into the skull under general anaesthesia. After several weeks of osseo-integration, a titanium top part was screwed onto the baseplate in a simple procedure. The headposts for the mice have been described in ref. 84. Briefly, the animal was placed under isoflurane anaesthesia, shaved and given local analgesia on the top of the head. An incision was made and the skin on top of the cranium was removed, before the cranium was cleaned and the custom milled titanium head plate was attached using dental cement.

### Experimental setup

Experiments were carried out in a darkened room (mice) or electrically shielded booth (monkeys). The animals were in the centre of a 120 cm diameter spherical dome extending to 250 deg visual angle. The headfixed mice were positioned on a styrofoam spherical treadmill; the headfixed monkeys were seated in a monkey chair and could spin a 12 cm diameter trackball with their hands. Movements of the spherical treadmill and trackball allowed the animals to traverse a virtual reality (VR) environment projected on the inside of the dome by means of a spherical mirror. Projecting the VR environment on a dome surrounding the animals enabled both their central and peripheral view to be covered, thereby providing an immersive and realistic VR environment. The VR environment was created using DomeVR, our custom-made toolbox combining photorealistic graphics rendered with Unreal Engine 4, with high timing precision required for neuroscience experiments[53].

### Experimental paradigm

Mice and monkeys were required to distinguish two natural shapes at equal distance in front of them, amidst a grassy field with a blue sky above and mountains in the background (Fig. 1A). The two shapes emerged out of a central shape which was either right at the starting position (for monkeys) or a short distance in front (for mice). A virtual collision with the correct shape yielded a reward ('hit'), whereas the incorrect shape yielded no reward ('wrong'), and no collision with either shape also yielded no reward ('miss') (2AFC paradigm). Rewards were drops of diluted juice for the monkeys and drops of vanilla soy milk for the mice. For the monkeys, the shapes varied smoothly between a non-rewarded, textured square and a rewarded triangle (monkey K) or between a rewarded, jagged and a non-rewarded, hour-glass shaped leaf (monkey C). On each trial, a blend between the two

shapes was shown alongside the exact middle blend ('reference shape'). For the mice, the shapes and their reward contingencies were the same as for monkey C.

Monkey data were recorded in 7 sessions for monkey C, 11 sessions for monkey K. Each session lasted about one hour, during which the monkeys completed $1208 \pm 186$ and $991 \pm 492$ trials at 67 and 77 percent correct (monkeys C and K, respectively). The monkeys were both fully trained on handling the trackball to move through the VR environment, as well as the VR task. Mouse data were recorded in (12, 4, 6, 3, 2, 2) sessions for mice (*001, 003, 004, 005, 012, 013*), respectively. Each session lasted about one hour, during which the mice completed $(280 \pm 103, 514 \pm 70, 573 \pm 112, 246 \pm 87, 462 \pm 8, 394 \pm 87)$ trials at (59, 54, 60, 77, 45, 63) percent correct (same mice ordering as before). Following the headpost surgery, the mice were handled for 5 days to reduce experimental anxiety due to head fixation and interaction with the experimenter, before behavioural training began. Behavioural training in the experimental setup at initial stages lasted between 3-5 sessions, before final data collection began, which lasted up to 30 sessions.

## Behavioural tracking

We recorded videos of the monkeys' and mice' faces during the tasks at 60 Hz using Basler acA640-121gm infra-red cameras with a modified version of PylonRecorder2 software (https://gitlab.mpcdf.mpg.de/mpibr/scic/pylonrecorder/PylonRecorder2). Additionally, in the monkeys, eye movements were recorded at 500 Hz using a Grasshopper3 infra-red camera and the free eye tracking software *iRecHS2*[85] and synchronized with DomeVR[53].

## Facial key point extraction

To extract facial key points from the videos, we used markerless pose estimation on them, as implemented in DeepLabCut[16,86]. For mice, features were extracted from videos of the left side of the face using our own model to identify key points such as the coordinates of the eye, whisker pad and nose. For mouse pupillometry, we used the eye coordinates from the face model to crop the video to include the entire left eye and ran it through a refined model based on the 'mouse pupil vclose' Animal Zoo model (provided by Jim McBurney-Lin at the University of California Riverside, USA) included with DeepLabCut. The output of the pupil model was 8 points covering the circumference of the mouse pupil, that were then used to calculate pupil and eye summary statistics.

For the macaque facial key points, we used the pre-trained 'primate face' model from the DeepLabCut Animal Zoo (provided by Claire Witham at the Centre for Macaques, MRC Harwell, UK) and extended it with additional points on the lips to capture more precise mouth movement than in the original model. All models were further trained and refined to achieve a detection error of less than 2 pixels per tracked key point in all conditions. The macaque raw pupil size recorded by the eye-tracker was Z-scored over time within the training data set.

To synchronise the video timing with events in the virtual reality environment, we used 32 ms long infra-red flashes emitted from an LED mounted near the camera lens. These flashes were then extracted from the face videos to be used as timestamps for synchronisation with DomeVR. Five consecutive flashes indicated the start of a behavioural session; a single flash indicated the start of a trial.

## Reaction time

In our VR setting, where animals move towards one of two stimuli rather than pressing a button or lever, or making an eye movement, we define the reaction time (RT) as the time point of the initial substantial movement directed towards either stimulus. While determining this time point, it is crucial to distinguish between stimulus-related movements and minor positional adjustments. We specifically focus on the first deviation in lateral movement, while excluding forward movement due to its susceptibility to random movements and its task irrelevance.

To calculate the RT, we use a sliding window linear regression approach, incorporating a time decay mechanism. This approach enables us to detect non-linearity by examining the coefficient of determination ($R^2$) for each window. A low $R^2$ value indicates that the data deviate from linearity, and such a deviation can be interpreted as a deviation in lateral movement.

First we compute a linear regression on the time series of lateral VR movement for adjacent sliding windows $i$ and $j$ of a given size ($n_w$). Then, $R_i^2$ (i.e., $R^2$ for window $i$) is calculated as:

$$R_i^2 = 1 - \frac{\sum_{j=1}^{n_w}\left(l_j - \hat{l}_j^i\right)^2}{\sum_{j=1}^{n_w}\left(l_j - \bar{l}\right)^2} \tag{2}$$

where $l_j$ is the $j^{th}$ element of the lateral movement observed in the second window, $\hat{l}_j^i$ is the corresponding predicted lateral movement value (based on window $i$) and $\bar{l}$ is the mean lateral movement within the second window. As a result, we get an array of $R^2$ values: $\mathbf{R}^2 = [R_1^2, \ldots, R_n^2]$.

Subsequently, we reverse the sign of the $-\mathbf{R}^2$ array and detect its local maxima. For this, we resort to the definition of extreme points (we have a univariate function in this case):

$$L = \underset{w}{\mathrm{argmax}}\left[\frac{d^2 r(w)}{dw^2}\right] \tag{3}$$

where we have simplified the notation, using $-\mathbf{R}^2 \equiv r(w)$. Once we have found the local maxima ($L$), we further require that they have a minimum prominence ($\lambda$). Prominence is a measure of the significance of a peak by comparing the peak to its surroundings:

$$\lambda_i = r(w_0) - \max\left[r(b_{l,i}), r(b_{r,i})\right], \quad \begin{aligned} b_{l,i} &= \underset{j \in [0, L_0]}{\mathrm{argmin}}\left[r(w_j)\right], \\ b_{r,i} &= \underset{j \in [L_0, n-1]}{\mathrm{argmin}}\left[r(w_j)\right] \end{aligned} \tag{4}$$

where $r(w_0)$ is $-R^2$ at $L_0$ and $b_l$ and $b_r$ are the arrays of left and right bases of the peaks; we are making use of the notation by which $r(w) \equiv -\mathbf{R}^2$.

For each peak in $r(w)$, we calculate the prominence and discard the ones that are below a given threshold ($\lambda_0$). The particular value for this threshold was not critical for the overall performance of the algorithm. For the sake of stability, we use multiple window sizes (100, 150, 200 and 250 ms) and combine the results in the following way. For each window $k$, we have an array of candidate points ($\mathbf{x}_{cand}^k$). Then, we create a vector of weights ($\mathbf{w}_k \in \mathbb{R}^n$) that have a value equal to a Gaussian distribution centred around each candidate point of each window. Mathematically:

$$\mathbf{w}_k(x) = \begin{cases} \mathcal{N}(x - x_{cand}^k, \sigma) & \text{if } x \in \mathcal{B}_{cand}^k \\ 0 & \text{otherwise} \end{cases}$$

where $\mathcal{B}_{cand}^k$ denotes the vicinity of each point in $x_{cand}$ for window $k$. Finally, the RT is given by:

$$RT = \underset{x}{\mathrm{argmax}}\left[\left(\sum_k \mathbf{w}_k(x)\right)/x\right] \tag{5}$$

Figure S1 shows the distribution of RTs split by trial outcome over sessions, for both species; Fig. S2 shows example paths and detected RTs for both species.

## Facial features

The extraction of the predictors for the MSLR model involves a multi-step process to go from continuous recording time (60 Hz for video data and 500 Hz for the macaque eye-tracker) to trial-based predictions.

First, we chose several points of interest on the animals' faces, which are then automatically identified and tracked over time using DeepLabCut[16]:

- Macaque: both ears, eyebrows, nostrils and lips (see Fig. S3A).
- Mouse: nose tip, left ear, left eye and median whiskers location (since we have a side view of the face, see Fig. S3B).

Once the data streams were aligned, we computed the median location $(x, y)$ of each facial point over the 250 ms window before the stimuli appeared on the dome. This time window was chosen to make sure that all of the facial expressions of the animals are due to internally generated processing, rather than stimulus processing. Different window sizes (particularly: 200, 300 and 500 ms) did not yield any qualitative difference. In addition to the median location, we also computed the total velocity of each facial point.

For both species, we further computed the median pupil size over the same time window. Pupil size is a well-known indicator of arousal and cognitive load, and thus provides valuable information about the internal state of the animal.

This resulted in a set of data points for each trial, corresponding to the median vertical and horizontal location, and total velocity of each of the facial features. These data points serve as the predictors for the MSLR model.

## Markov-switching linear regression

Markov-Switching Linear Regression (MSLR) models, which we ran using *Dynamax*[55], are a powerful tool for modelling time series data that exhibit regime-switching behaviour, where the underlying dynamics of the system change over time.

The MSLR model is defined by a set of linear regressions, each associated with a particular state of a discrete Markov chain. The state of the Markov chain determines which sets of weights and biases predict the evolution of the observed data at each time step. The transitions between states are governed by the transition probabilities of the Markov chain, which are learned from the data.

Formally, an MSLR model can be described as follows. If $S$ is the total number of latent (discrete) states of a Markov process, at each time step $t$, a given state $z_t (\in \{0, 1, ..., S\})$ will follow a Markovian evolution such that:

$$P(z_{t+1} = j | z_t = i) = \pi_{ij} \tag{6}$$

As these are stochastic matrices, $\pi_{ij} \in [0, 1]$.

Let the $M-$ dimensional input time series at time $t$ be denoted by $x_t (\in \mathbb{R}^M)$. Let the $N-$ dimensional output time series at time $t$ be denoted by $y_t (\in \mathbb{R}^N)$. Then, in the case of a MSLR, the discrete latent variable at time $t$ ($z_t$), will dictate which emission weights ($W \in \mathbb{R}^{N \times M}$) and emission biases ($b_s \in \mathbb{R}^N$) we will use to predict the outputs (emissions) based on the inputs (predictors). Moreover, an emission covariance matrix ($\Sigma_s \in \mathbb{R}_{\succcurlyeq 0}^{N \times N}$) will also have to be learnt. Explicitly, at time $t$, the emission distribution in this model is given by:

$$p(y_t | z_t, x_t, \theta) = \mathcal{N}(y_t | W_{z_t} x_t + b_{z_t}, \Sigma_{z_t}) \tag{7}$$

Therefore, the problem of fitting this model amounts to finding the set of emission parameters denoted by:

$$\theta = \left\{ (W_s, b_s, \pi_s, \Sigma_s) \right\}_{s=1}^{S} \tag{8}$$

In other words, the aim is to find the weights ($W_s$) and biases ($b_s$) for the linear regressions and the transition $\pi_s$ and covariance $\Sigma_s$ matrices for the Markov process.

In our case, the discrete latent variable ($z_t$) represents the internal state of the animal at trial $t$, which is inferred from the facial features ($x_t$) extracted using DeepLabCut[16] and the observation ($y_t$) that represents the RT of the animal. We trained the MSLR model using the Expectation-Maximization (EM) algorithm[87], which iteratively computes the probability over latent states given the data and updates the model parameters to maximize the likelihood of the observed data. For further details, we refer the reader to ref. 88. We iterated the EM algorithm for 50 times, for all models. We initialized the model parameters using a normal distribution for weights and biases and we used the identity matrix as the initial covariance matrix for the emissions. We assumed a Dirichlet prior for the transition matrix. We repeated this process 10 times to increase confidence that we got the optimum value for each combination of parameters.

## Training and inference

We used an 80: 20 ratio for train-test splitting and performed hyperparameter optimization by cross-validating the training set only (see Methods - Model tuning for details on CV and model selection). For each species, we concatenated the training sets of all sessions, with forced transitions in between the sessions (setting predictors and emissions to 0 for 50 consecutive trials), so that state probabilities are reset. Then, after optimizing each model, we performed inference on each held out test set (separately per session). We decided to take this approach for various reasons:

- Model generalization: as the model learns from potentially different faces, it is likely that it can pick up on common information between them.
- Model interpretability: given that we do not update the model parameters at the inference step, all internal states have the same meaning over subjects and, thus, are directly comparable.
- Better convergence: increasing the number of training samples (i.e. concatenating sessions as opposed to training a different model per session) allows the model to have more data to learn from.

All of the results in the main text, unless otherwise stated, are for held out data.

## Model tuning

For the model we described in *Markov-Switching Linear Regression*, there are several parameters that can be tuned to explain the data better. We assumed a Dirichlet prior distribution, as it is the conjugate prior distribution of the categorical distribution. The concentration parameter ($\alpha$) controls the relative density or sparseness of the resulting transition matrix. We decided to explore the influence of changing the maximum number of internal states ($S$) and to add sticky transitions to the Markov process (a self-bias term in the transition matrix $\pi$, making states taking longer to transition to a different one); the parameter that controls this is the stickiness ($\beta$), and to vary the transition matrix sparsity (concentration).

In order to balance model performance with scientific insights, we took a hybrid approach. We increased the number of internal states in a greedy way, to show that the error saturates and that there are diminishing returns when increasing model complexity. On the other hand, for a given number of states, we optimized the two free parameters of the Markov process ($\alpha$ and $\beta$).

For the sake of efficiency, we used Optuna[89], a flexible framework to implement Bayesian optimization. In Table 1 we report the relevant quantities for this process.

To select the best combination of parameters, we performed 5-fold Time-Blocked Cross-Validation[90].

**Table 1 | Parameter values for the Bayesian parameter optimization procedure**

| Parameter | Range | Fitted Values |
|---|---|---|
| Concentration ($\alpha$) | [0, 100] | $\alpha_{mac}$ = 19.3, $\alpha_{mou}$ = 34.3 |
| Stickiness ($\beta$) | [0, 100] | $\beta_{mac}$ = 45.7, $\beta_{mou}$ = 42.0 |
| Procedure | Setting | Notes |
| Sampler | CMA-ES[98] | – |
| Objective Function | $R^2$ | Maximized |
| Number of Searches | 100 | – |

The fitted values are shown for macaque (mac) and mouse (mou) scenarios, independently explored for each number of internal states in the HMM.

We ended up selecting these values, after cross-validation: for macaques: $\alpha$ = 19.315, $\beta$ = 45.705; for mice: $\alpha$ = 34.312, $\beta$ = 42.029. These values explain the overall tendency we find in the main paper: macaques have a more block-like behaviour (sparser and more diagonally-dominant transition matrix), whereas mice transition more and in between more states (denser and less diagonally-dominant transition matrix).

### Synthetic data and ground truth states

In order to validate the retrieval of states when we do not have access to ground truth ones, we generated a time series of ground truth emissions and states based on the given inputs (by using the same input data as in the main text). To this end, we trained an MSLR model with a known given number of states and sampled some emissions and states sequence from it. We aimed to recover the appropriate number of states with the correct temporal sequence, and to correctly predict the emissions.

In Fig. S4A, we show that the model's $R^2$ saturates at the ground-truth number of states (dashed vertical line). Nevertheless, we took the best performing number of states ($n_s$ = 10) and cluster (through a hierarchical clustering algorithm, details below) these states based on their predicted emissions, and compare their similarities against a surrogate distribution (constructed via 500 shufflings of one of the input predictions, thus destroying any temporal structure). For those within or below a significance range (chosen to be the 90th–99th percentile of the surrogate distribution), we group them together. At the end of this process, we end up selecting 5 states (thus recovering the ground-truth $n_s$), see Fig. S4B.

For the selected number of states, we fit a MSLR using the EM algorithm for 50 steps (same as in the main paper) and plot the state sequence (true vs inferred), see Fig. S4C. To quantify the observation that the inferred states closely match the ground-truth ones, we computed their correlation (one-hot encoding the true and predicted state sequences), showing a very high value for each value across the diagonal (and only for that one), see Fig. S4D. There is a very close match ($\rho(s_{true}, s_{pred}) > 0.7$) between the true and inferred states, given that the $99^{th}$ percentile of the surrogate correlation distribution was 0.12. We mask all values below this surrogate threshold.

### Hierarchical clustering

We clustered $n$ states based on their emission distributions. In order to quantify the pairwise distribution similarity, we used the Jensen-Shannon Distance, so we first needed to turn each state-specific RT emissions distribution into a histogram. To that end, we used the Freedman-Diaconis rule to automatically select the optimal number of bins.

### Face separability

We computed a "face separability" measure, which quantifies how dissimilar the different distributions of facial features are when split over states.

Mathematically, this measure is defined as:

$$\Omega = \max(A, 1 - A) \qquad (9)$$

where A is Vargha-Delaney's A-statistic (also known as measure of stochastic superiority), an effect size derived from the Mann-Whitney U-test[91]. We decided to use $\Omega$ because it is especially interpretable. As it is related to the U-statistic, it can be thought of as the probability of a randomly selected point from one distribution being higher than another randomly selected point from the other distribution. This measure is bounded between 0.5 and 1. If there is no overlap, $\Omega$ = 1. In this extreme case, one distribution would have complete stochastic dominance over the other. If $X$ and $Y$ are completely overlapping, $\Omega$ = 0.5. Intermediate cases reflect the degree of dominance one distribution has over the other. The more its value deviates from 0.5, the less overlapping the distributions are.

We have shaded regions according to the common interpretation[91]:

- Small separability (grey): $0.56 < \Omega < 0.64$.
- Medium separability (orange): $0.64 < \Omega < 0.71$
- High separability (red): $\Omega \geq 0.71$

For example, for the first column in the macaque row (Fig. S5, State A vs B), taking pupil size (bottom row) as an example, this metric quantifies how differently distributed the pupil size in state A vs state B is over sessions. Its median value is around 0.72, meaning high separability. This implies that this feature is very different across these two states, whereas the vertical left ear position (lEar [y] in the plot, top distribution) is much less useful to disambiguate between these two states. Comparing the separability (across features) between the various state combinations, two main results emerge:

- There are some states whose faces are similar to each other (Macaque: State C vs A; Mouse: State B vs C). This is in line with what we are showing in Fig. 5A in the main text.
- No single feature is useful to disambiguate between states. This points back to one of our conclusions in the main paper: when accounting for internal states, the field is currently discarding potentially useful information by only relying on a single feature (most often, pupil size).

### ARHMM

As we wanted to ensure that facial features were indeed informative of reaction times (RTs) beyond what is to be expected by the RT auto-correlation structure, we implemented an Auto-Regressive Hidden Markov Model (ARHMM). In this case, we used the same pipeline as we detailed in the previous sections, but substituted the facial features at the current trial $t$ for the RT of the previous trial ($t - 1$). As it can be seen from Fig. S6, the facial features model outperforms the ARHMM for all states, for both species.

### Task performance and internal states

We were interested in investigating whether the inferred internal states were correlated with task performance, even though the model had not been trained on such information. We therefore used the predicted single-trial state probabilities to decode choice, using a simple Logistic Regression model, with an $L2$ penalty term. After verifying that the model does indeed classify outcomes beyond chance level (Fig. S7), we took the weight of each state as a proxy for how related it was to each outcome.

### Software and Tools

All analyses were performed in Python. For data handling and numerical computations, we used NumPy[92] and Pandas[93]. Statistical analyses and signal processing were conducted using SciPy[94] and

Scikit-learn[95]. Visualizations were created using Matplotlib[96] and Seaborn[97].

## Reporting summary

Further information on research design is available in the Nature Portfolio Reporting Summary linked to this article.

## Data availability

The data sets analysed during the current study, containing the quantified facial features over time as outputted by our DeepLabCut models, are publicly available at https://github.com/atlaie/thoughtful-faces/.

## Code availability

All the relevant scripts are publicly available at https://doi.org/10.5281/zenodo.14850103.

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

## Acknowledgements

We thank Carlos Wert Carvajal for useful input on earlier versions of the manuscript, and we thank Victor Geadah, Iris Stone and Matthew Creamer for their valuable comments on model training. A.T. acknowledges support from the Margarita Salas Fellowship (Spanish Ministry of Economy) and from the Add-On Fellowship for Interdisciplinary Life Sciences (Joachim Herz Stiftung). This work was funded by the Max Planck Society.

## Author contributions

Alejandro Tlaie: Project conceptualization, data analysis, software development, manuscript writing and editing, funding acquisition; Muad Y. Abd el Hay: Data curation and analysis, software development, manuscript writing and editing; Berkutay Mert: Data analysis. Robert Taylor: Experimental setup, animal training, data acquisition, manuscript editing; Pierre-Antoine Ferracci: Experimental setup, animal training, data acquisition; Katharine Shapcott: Project conceptualization, software development, manuscript editing; Mina Glukhova: Software development, data acquisition; Jonathan W. Pillow: Project conceptualization, supervision, data analysis, manuscript editing; Martha N. Havenith: Project conceptualization, supervision, manuscript writing and editing, funding acquisition; Marieke L. Schölvinck: Project conceptualization, supervision, manuscript writing and editing, funding acquisition.

## Funding

## Competing interests

The authors declare no competing interests.
