## [Transparent Peer Review file · Nature Communications]

Inferring internal states across mice and monkeys using facial features

Corresponding Author: Dr Alejandro Tlaie

Version 1:

Reviewer comments:

Reviewer #1

(Remarks to the Author)

Thanks very much to the authors for addressing my concerns rigorously with new analyses, and better explaining the analyses. It seems clear that these behavioral/internal states are distinct. One major concern I still have is that the authors have not shown that the model can work on a held-out mouse or monkey, this is necessary to claim that these states are generalizable. Can the authors at least show that states A,B,C map onto for example low,high,low reaction times like in Fig S19 in a held-out mouse (and/or hit,miss,wrong) (e.g. re-fit the model with n-1 mice and test on 1 mouse for all the permutations)?

One minor concern, Fig S13: the authors claim that removing pupil area does not decrease model predictivity of reaction time – they say “This speaks against the usual perspective by which pupil size is paramount in inferring internal states”, however, isn’t this alternatively evidence that the tracked keypoints are all potentially correlated? To make their stated claim, the authors need to hold out various keypoints and show the change in R-squared, and show that one of them besides pupil causes the most decrease in R-squared.

(Remarks on code availability)

Reviewer #2

(Remarks to the Author)

I thank the authors for the thorough revision of the paper. Overall, the authors addressed many of my original concerns, and I appreciate the efforts both in the text and the figures. I agree with the authors’ assessment that the paper has already substantially improved.

The cross-species comparisons became clearer, and the updates to both intro and discussion for positioning in the literature are helpful. Conclusions and claims are much better supported now compared to the original version due to updates to main and supplementary figures. My main remaining concern is about further strengthening the motivation of the state selection and demonstrate that the observed states and their interpretation are robust. Besides this, more statistical analysis could be added to back claims about the interpretation of the states. Other remarks are added below and only require minor changes.

In my first review, I noted, “the paper story depends a lot on the selection of appropriate states, which is visualized in Fig 2. I am not fully convinced of the methodology and depth of the analysis here, and think that further analysis could strengthen the claims of the paper.”

The authors already made great efforts to improve the rigour of this analysis, but a few concerns remain. I still think that the validation of the selection scheme and interpretation of states needs another revision. In particular, there is a mismatch between the synthetic validation experiment and the real data, and additional clarification is needed on the finite difference selection scheme.

My summary of the current protocol:

- The authors use synthetic data to verify and justify their model, which is good and I understood the data generation for that given the clarification in the rebuttal. In Fig. S4, a dataset with 5 ground truth states is generated and used to validate the model. To recover the correct number of states, the R2 is considered (Fig. S4A). The model with 5 states has the highest R2 and is then examined for further analysis in Fig. S4C-D.
- The identical metric (?) for the real data is depicted in Fig 2A. As the authors note, if the same selection scheme as for the synthetic validation would have been applied, the resulting number of states is much higher than what was considered in the main paper (10 and 8, cf. Fig. S4 caption, but see my question below). On the synthetic validation, R2 peaks for the correct number of states, on the real data, it keeps improving.
- The actual metric to choose from on the real data, is in Fig. S14, which is the finite differences in cv-R2.

Given this, I have the following questions for the authors:

(1) I do not fully understand why the R2 in Figure S4A declines after 5 states, but keeps improving in Fig 2A. Is this an overfitting effect for the synthetic data we do not see for the real data? If yes, did you consider checking the R2 for both the train and validation set in the synthetic and real data?

My general suggestion would be to improve the synthetic data experiment, otherwise the connection between the synthetic data experiment and its value for the computational analysis is not clear to me. Specifically, Figure S4 should use the same selection protocol (incl. cross-validation methodology, metrics, etc.) as the real data experiments. This would include plotting the data in a similar way as for the real data (the figures should be visually similar e.g. to Fig 2A, Fig 4A, Fig S14). Right now, it seems like using finite differences (as in S14 for the real data) would not yield the correct 5 states on the synthetic dataset.

(2) The match between the numbers reported in Fig S14 and Fig 2A or Fig S6 is not clear. In the figures depicting absolute cv-R2, the performance of a model with 2 states is better than a single state (that is one motivation for picking a model with multiple states). Yet, for both the mouse and monkey, Fig S14 shows a decline around -.5 to -.3 when introducing the 2nd state. In Fig 2A, the decline should only happen for the 3rd and 5th state (which is consistent with S14 again). Could you clarify the metric used for S14 and the relation to Figure 2A, in particular if the difference plotted is relative (as the caption suggests) or absolute (as the figure y label suggests)? Apologies for missing this in the first review. Could you also clarify how $n=10$ and $n=8$ in Fig S10 was chosen? In Fig 2A, it looks like performance keeps increasing also beyond these values?

Besides the synthetic validation, I have the following additional questions about state selection and the interpretation of states:

(3) For the monkey, there is a state only associated to "Wrong" (state A) which does not exist in the mouse, and in the mouse there is a state associated to "Hit" only (state A) which is not present in the monkey. This might be a potential confounder of some of the following analysis, and is not really addressed. Only the states C (positively correlated to "Hit & Wrong") and the state D (monkey), B (mouse) ("Miss") match well across species. In my sense, this "mixed selectivity" is not properly addressed in the current version of the manuscript, and seems to be an important difference between the species. I would be interested in further clarification.

(4) In the new analysis figure S10 where 10 and 8 states are selected, it also seems like the monkey has much clearer defined states (4 states, all others are redundant from visual inspection), while the mouse uses a broader repertoire (H, C, D, E, F seem to be distinct). It would be interesting to consider a data-driven way of "collapsing" the states across species, e.g. by clustering or another suitable technique. This could also be another validation that the finally selected states are robust (if the collapsed states resembled the 3/4 states originally fitted).

(5) II. 672-676R ("mapped robustly onto behavioral trial outcomes"): This statement implies that under changes in parameters of the underlying statistical model or the data, the mapping would stay similar/the same. I think that such an analysis was not conducted. Given my comments about state selection, this statement should either be down-tuned (e.g., drop mention of "robust"), or a robustness analysis should be conducted, e.g. by running the experimental protocol on subselections of the data, followed by model fitting, state selection by the final metric (see 1,2) and summarizing the results. This might also be an addition to the synthetic validation experiment.

A few additional comments, in chronological order as they appear in the manuscript:

(6) L. 432L, with ref to Fig S10: This statement could be more quantitative, what is the measure of similarity here?

(7) It would strengthen the claims if the statements in II. 485L-575L, II. 563R-575R were backed up by statistical tests, especially when explicit statements about significance (I. 526L) are made. In general, the paper would benefit from more statistical analysis of the reported results.

(8) II. 635-637L, where is the "minimal shaped by task history" investigated?

(9) II 787-799R reference "studies of internal states ...", "... compared to previous research ...", but no references to literature are given.

(10) There are multiple references of overfitting or avoiding overfitting (e.g. in S10 caption), but actually no analysis that

investigates this statement (e.g. in terms of a line plot where the parameter of interest changes along the x-axis, and the selection metric is plotted along the y-axis for evaluation on the train and validation set). If the claims about overfitting are correct, in S10 we should see that the train performance continues to improve, but the validation performance starts degrading once the # states become too high. Is this the case?

Minor comments:

- l. 23, foundational, l. 29 fundamental -> would pick one, fundamental seems more appropriate?
- Fig S14, typo on y label: "R" instead of "R2" as in the caption.
- What is the x-axis in S17, the norm of the feature weight?
- Caption S4, "converges to the ground truth value" -> this does not seem correct, it converges to a slightly lower value in S4B. Maybe "approaches" is a better choice of words, "converges" implies that it actually converges towards the value in the limit of many EM steps, that does not seem to be the case.
- Colormap of Fig S4D could match the colormap of the state mapping in Fig 2A.
- Is the red/green colormap used in the figure colorblind friendly? An additional consideration for accessibility might be to also add numbers to the plot in Fig 2A (and related plots).

Replies to the rebuttal comments:

Below are all replies from the rebuttal document that need further discussion. All other comments have been addressed, thanks again for the numerous updates!

> We already depict the input device for the monkey in Fig. 1B (top row, spherical trackball), maybe the reviewer mistook this for a movie instead of a monkey hand and trackball schematic?

[minor] You are right and I can actually see this in the digital PDF version, but strangely, in the printed version of the manuscript I used for reviewing, this (and only this) element is missing (also in the revised version; might be an issue with my printer, but potentially good to double check).

> We agree with the reviewer that it was difficult to precisely compare the plots visually, and have now added rounded numbers into the cells of the matrices as suggested.

Thanks for this addition. Could you add these numbers also in Fig. S16? The number of digits might still be unified (e.g. in X.XX% format, or by keeping the # of significant digits or total digits the same?)

> We agree with the reviewer that this statement needed to be demonstrated more directly. To this end we have added the new Supp. Figs. S5 and S13. In Supp. Fig. S5, we show that the separability of facial features between different states is uniformly strong, implying that there is no individual feature that is particularly informative in distinguishing between two states. Moreover, the most informative features vary depending on which states are compared. The revised Fig. 5C in the paper demonstrates a similar point, showing that each facial feature contributes varying weights to different states. In addition, we also show that when a particular feature that has been highlighted a lot in previous work, namely pupil size, is excluded from the analysis, model performance remains unaltered (new Supp. Fig. S13). This again indicates that information is distributed across multiple facial features.

I really like this addition. When looking at Fig. S5, I agree with the interpretation. Figure S13 is also a good control for the pupil size result. Minor suggestion: I would consider adding an early reference to Fig. S13 into the text in lines 1318-1327 to address this directly?

Regarding the A-statistic, is it possible to add the corresponding statistical test (visually it looks like only the pupil size for A vs. B might be significant)?

> We are very sorry, but we would need a more detailed description of this suggestion to put it into action, as we are not sure which histograms and state distributions are being referred to here.

Apologies for being unclear. This has been already addressed with the addition made to Figure 3. I was interested in seeing an overview of how many trials are associated with each state. The panel "#trials/session vs. State" in Fig 3B is sufficient for that.

(Remarks on code availability)

- The python version used is not specified. I tried to install the code with python 3.8,3.9,3.10,3.11 and 3.12, but ran into dependency issues in all cases; one common problem was the installation of the "dynamax" package. If exact dependencies are required, a conda env or docker container could clarify the exact build setup. Otherwise, clarifying the python version plus updating the requirements to successfully solve all dependencies would be helpful. Apologies if I missed something obvious.
- The code is otherwise mostly clear and commented, and the structure is clear.
- The README needs to be revised. The order of operations (e.g. installing the dependencies before cloning the repository) is not ideal. Also, some of the code files have been moved into folders, but this is not reflected in the example commands provided.
- Jupyter notebooks are added to the repository, but I think not updated according to the revision and new figures. It would be

great if the authors could ensure that the code is in sync with the current version of the manuscript.

- It would be a great addition if the Jupyter notebooks were better commented. Minimally, labels to the figures could be added (optionally, along with the caption text), it is otherwise hard to navigate the document. Another suggestion would be to split the plotting notebook by figure (optional!).

Note, I did not conduct a deeper code review at this stage, and did not actively reproduce any results.

Version 2:

Reviewer comments:

Reviewer #1

(Remarks to the Author)

Thank you very much for performing the leave-one-out analysis, it is clear the model performs well. Perhaps it could be added to Figure 2A as an additional whisker on the plot, it is a bit unfortunate to leave an important analysis for Figure S22. And thank you for the additional pupil analysis, it is an interesting finding that is well-supported.

(Remarks on code availability)

Reviewer #2

(Remarks to the Author)

I thank the authors for the replies to my comments and the additional analysis. I especially appreciate the new figures and analysis aimed to address the comments about the synthetic data setup.

In summary, my biggest concern in the previous review was the mismatch in the validation of the main analysis method on synthetic vs. real data. This synthetic validation used a different model selection protocol (for the number of states) than what the authors used on the real dataset.

The authors now revised Figure S4A to use a validation protocol in which internal state number is by a hierarchical clustering method; on the synthetic data, selecting the best performing models according to cross-validated R2 then over clusters the number of states, selecting 10 instead of 5. This is followed by a hierarchical clustering step to group states together, and uses a shuffle control to determine the number of appropriate states. This selection mechanism is able to discover the correct number of ground truth states with acceptable correlation to the ground truth states.

The authors then re-analysed (in the supplement, not the main paper) their dataset using this selection protocol and show support for 2-4 states (monkey) or 2-4 states (mice), and the states reported in the paper would fall into these ranges. It remains debatable if the resulting states in Fig. S12 (state 1, Wrong) and Fig. 4A (state A, Wrong) are fully comparable/well recovered: The former has a median reaction time around ~.2s, the latter around ~.5s (eyeballed from the plots). This is partially covered by statements in the main paper, which make less strong claims about this "Wrong" state.

In any case, the additions to the data analysis in my opinion sufficiently support findings resulting from the state selection, and where deviating interpretations are possible. I thank the authors for the additional time invested in better contextualising their analysis results.

Further minor comments, mostly related to these additions:

- l. 1212 "Figure S4A illustrates the input data" -> seems to be a typo due to reorganisation of plots, there is no data referenced in that figure
- l. 1214 "does peak at the ground truth one" -> "does converge to the ground truth one" ?
- l. 1215 "for some example trials" -> why not for all?
- l. 1219 "there is an almost perfect match ... > .9" -> the figure shows ranges from .73 to .91, this statement seems to be inaccurate.
- Fig S4D: What about the off-diagonal elements? Right now, it looks like all these correlations would be 0, but that does not appear plausible.
- Fig S4, caption: "approaches to" / "approaches"
- l. 1245-1247 these statements reference "significant" or "not significant" results, but do not state a p-value and the used test. The referenced figure also contains no details about this.

(Remarks on code availability)

I thank the authors for revising and improving the code. I did not attempt to re-run the code, but noticed a few minor remaining issues, e.g. in the README the path to the requirements file is not exactly matching, or there are duplicate files in the repo, like "mouse_MSLR_final copy.py" vs. "mouse_MSLR_final.py". A final cleanup and check would be good.

Response to Reviews:

Thoughtful faces: inferring internal states across species using facial features

Alejandro Tlaie, Muad Y. Abd El Hay, Berkutay Mert, Robert Taylor, Pierre-Antoine Ferracci, Katharine Shapcott, Mina Glukhova, Jonathan W Pillow, Martha Havenith, Marieke Schölvinck

General remarks

We thank the reviewers for their valuable comments, which have substantially improved the manuscript. Please see our point-by-point replies below.

Response to Reviewers

Reviewer #1

Thanks very much to the authors for addressing my concerns rigorously with new analyses, and better explaining the analyses. It seems clear that these behavioral/internal states are distinct. One major concern I still have is that the authors have not shown that the model can work on a held-out mouse or monkey, this is necessary to claim that these states are generalizable. Can the authors at least show that states A,B,C map onto for example low,high,low reaction times like in Fig S19 in a held-out mouse (and/or hit, miss, wrong) (e.g. re-fit the model with n-1 mice and test on 1 mouse for all the permutations)?

We agree with the reviewer that this is an important point and thank them for asking about this. Following their suggestion, we have now repeatedly retrained the model on data from 5 mice, and applied it to data from 1 held-out mouse. The results are shown in the new Fig. S22 . As one can see, model performance on held-out individuals was consistently high - in fact, it was indistinguishable from overall model performance for all mice. In addition, states were distributed across different ranges of reaction times in highly similar ways across sessions as well as animals (see figure below), further confirming that the states we identified generalized well across animals, both in predictive performance and in their behavioural attributes.

One minor concern, Fig S13: the authors claim that removing pupil area does not decrease model predictivity of reaction time – they say “This speaks against the usual perspective by which pupil size is paramount in inferring internal states”, however, isn’t this alternatively evidence that the tracked keypoints are all potentially correlated? To make their stated claim, the authors need to hold out various keypoints and show the change in R-squared, and show that one of them besides pupil causes the most decrease in R-squared.

We thank the reviewer for pointing this out. We partially addressed this concern in Fig. S9, where we computed the Variance Inflation Factor. This analysis showed that very few of the facial features included in our models were significantly correlated (Left Eyebrow [x] for macaques, or Eye Movement for mice), which we removed prior to model fitting. This indicates that this is likely not the reason for the model's strong performance without the pupil size parameter.

To address this concern more fully, we have now added an analysis mapping the correlation of all behavioural variables with pupil size, and then removed the only variable that significantly correlated with pupil size (eye movement, for macaques). When we re-trained and tested the model after additionally removing this feature, we saw no significant drop in test performance. We have now included these analyses as the new Fig. S16.

Reviewer #2

I thank the authors for the thorough revision of the paper. Overall, the authors addressed many of my original concerns, and I appreciate the efforts both in the text and the figures. I agree with the authors' assessment that the paper has already substantially improved.

The cross-species comparisons became clearer, and the updates to both intro and discussion for positioning in the literature are helpful. Conclusions and claims are much better supported now compared to the original version due to updates to main and supplementary figures. My main remaining concern is about further strengthening the motivation of the state selection and demonstrate that the observed states and their interpretation are robust. Besides this, more statistical analysis could be added to back claims about the interpretation of the states. Other remarks are added below and only require minor changes.

We thank the reviewer for their encouraging and very helpful comments!

In my first review, I noted, "the paper story depends a lot on the selection of appropriate states, which is visualized in Fig 2. I am not fully convinced of the methodology and depth of the analysis here, and think that further analysis could strengthen the claims of the paper."

The authors already made great efforts to improve the rigour of this analysis, but a few concerns remain. I still think that the validation of the selection scheme and interpretation of states needs another revision. In particular, there is a mismatch between the synthetic validation experiment and the real data, and additional clarification is needed on the finite difference selection scheme.

My summary of the current protocol:

- The authors use synthetic data to verify and justify their model, which is good and I understood the data generation for that given the clarification in the rebuttal. In Fig. S4, a dataset with 5 ground truth states is generated and used to validate the model. To recover the correct number of states, the R2 is considered (Fig. S4A). The model with 5 states has the highest R2 and is then examined for further analysis in Fig. S4C-D.
- The identical metric (?) for the real data is depicted in Fig 2A. As the authors note, if the same selection scheme as for the synthetic validation would have been applied, the resulting number of states is much higher than what was considered in the main paper (10 and 8, cf. Fig. S4 caption, but see my question below). On the synthetic validation, R2 peaks for the correct number of states, on the real data, it keeps improving.
- The actual metric to choose from on the real data, is in Fig. S14, which is the finite differences in cv-R2.

Given this, I have the following questions for the authors:

(1) I do not fully understand why the R2 in Figure S4A declines after 5 states, but keeps improving in Fig 2A. Is this an overfitting effect for the synthetic data we do not see for the real data? If yes, did you consider checking the R2 for both the train and validation set in the synthetic and real data?

My general suggestion would be to improve the synthetic data experiment, otherwise the connection between the synthetic data experiment and its value for the computational analysis is not clear to me. Specifically, Figure S4 should use the same selection protocol (incl. cross-validation methodology, metrics, etc.) as the real data experiments. This would include plotting the data in a similar way as for

the real data (the figures should be visually similar e.g. to Fig 2A, Fig 4A, Fig S14). Right now, it seems like using finite differences (as in S14 for the real data) would not yield the correct 5 states on the synthetic dataset.

We thank the reviewer for the very pertinent observations and questions. Based on these, we decided to change our synthetic analyses and thus refactor Fig. S4 entirely (also displayed here for convenience). Now, the pipeline that we use in the synthetic and empirical case are exactly mirrored.

Before, we were generating the synthetic data from learning the underlying distributions, without paying attention to the temporal dynamics (which are key in this case). In contrast, we are now generating the data from a ground-truth MSLR based on the inputs, with a known number of internal states ($n_s=5$). We sample from that ground-truth MSLR and construct a synthetic time series, which will be our prediction target.

We begin by computing the CV R^2 (same number of folds and repetitions) over internal states and select the best performing one ($n_s=10$ in the synthetic case, even if the ground-truth is $n_s=5$), see panel A. We then cluster (through a hierarchical clustering algorithm, details below) these states based on their predicted emissions, and compare their similarities against a surrogate distribution (constructed via 500 shufflings of one of the input predictions, thus destroying any temporal structure). For those within or below a significance range (chosen to be the 90th-99th percentile of the surrogate distribution), we group them together. At the end of this process, we end up selecting 5 states (thus recovering the ground-truth n_s), see panel B.

For the selected number of states, we fit a MSLR using the EM algorithm for 50 steps (same as in the main paper) and plot the state sequence (true vs inferred), see C. To quantify the observation that the inferred states closely match the ground-truth ones, we computed their correlation, showing a very high value for each value across the diagonal (and only for that one), see D.

As a technical side note: In order to quantify the pairwise distribution similarity, we have used the Jensen-Shannon Distance, so we have turned each state-specific RT emissions distribution into a histogram (using the Freedman-Diaconis rule to automatically select the optimal number of bins).

(2) The match between the numbers reported in Fig S14 and Fig 2A or Fig S6 is not clear. In the figures depicting absolute cv-R2, the performance of a model with 2 states is better than a single state (that is one motivation for picking a model with multiple states). Yet, for both the mouse and monkey, Fig S14 shows a decline around -.5 to -.3 when introducing the 2nd state. In Fig 2A, the decline should only happen for the 3rd and 5th state (which is consistent with S14 again). Could you clarify the metric used for S14 and the relation to Figure 2A, in particular if the difference plotted is relative (as the caption suggests) or absolute (as the figure y label suggests)? Apologies for missing this in the first review. Could you also clarify how $n=10$ and $n=8$ in Fig S10 was chosen? In Fig 2A, it looks like performance keeps increasing also beyond these values?

We thank the reviewer for investing the effort to give such thoughtful and constructive feedback, we believe it greatly improved our approach to how we are now selecting the number of states. The discrepancy they detected is due to the fact that we were computing finite differences between consecutive R^2 by pre-appending the mean of the curve. In other words, the mean R^2 across all n_s was subtracted from the first value. We appreciate that this approach adds confusion, and now set the first

ΔR^2 value (for $n_s=2$) to 0, as we are interested in how much we gain when we increase the number of states from 2 onwards. We have also changed the layout to barplots, as it makes the overall take-home message more clear (see the new panel A).

The number of states ($n_s=10$ and $n_s=8$ for monkeys and mice, respectively), was chosen because of them having the highest R^2 CV performance. This is now explained in Results section D, lines 398-406. Based on these state numbers, and just as in the case of our new synthetic data, we clustered the best performing number of states ($n_s=10$ and $n_s=8$, for macaques and mice respectively), using the same hierarchical clustering algorithm as in the synthetic case. This yields a possible choice of $n_s \in \{2, 3, 4\}$ for macaques and $n_s \in \{3, 4\}$ for mice.

Given that the strongest improvement in R^2 occurred for 3 states in mice and 4 in macaques, these two analyses together supported our choice of 4 and 3 states for macaques and mice respectively.

These results are now shown in the updated Fig. S11 (reproduced here for convenience).

(3) For the monkey, there is a state only associated to “Wrong” (state A) which does not exist in the mouse, and in the mouse there is a state associated to “Hit” only (state A) which is not present in the monkey. This might be a potential confounder of some of the following analysis, and is not really addressed. Only the states C (positively correlated to “Hit & Wrong”) and the state D (monkey), B (mouse) (“Miss”) match well across species. In my sense, this “mixed selectivity” is not properly addressed in the current version of the manuscript, and seems to be an important difference between the species. I would be interested in further clarification.

We thank the reviewer for highlighting these very interesting effects! We agree that the constellation of behavioural outcomes for different states opens up some intriguing interpretations regarding the similarities and differences in attentive processing for mice and monkeys. We would also like to point out that e.g. the differences between monkey state A and mouse state C are in fact quite small numerically, so that cross-species differences in these two states are quite likely to be quantitative rather than qualitative.

What’s more, since we only work with one value per state-outcome association, there is no obvious way to test such more subtle cross-species differences statistically. We have therefore so far refrained from mentioning some of the (in our view very plausible) comparisons the reviewer points out. What we can

say when putting together the results shown so far, as well as the new analyses shown in answer to points 4 and 7, is that in both species, states linked to miss trials seem to behave genuinely differently from both hit and wrong trials, while states associated with hit and wrong trials, respectively, seem to be more interchangeable, both in terms of their behavioural outcome probabilities, and the facial features associated with them. We expand on this in the Results section E, lines 451-461.

(4) In the new analysis figure S10 where 10 and 8 states are selected, it also seems like the monkey has much clearer defined states (4 states, all others are redundant from visual inspection), while the mouse uses a broader repertoire (H, C, D, E, F seem to be distinct). It would be interesting to consider a data-driven way of "collapsing" the states across species, e.g. by clustering or another suitable technique. This could also be another validation that the finally selected states are robust (if the collapsed states resembled the 3/4 states originally fitted).

We thank the reviewer for this suggestion and have now subjected the states shown in Fig. S10 to a hierarchical clustering algorithm, as discussed in more detail in Point 2 above. The figure below shows how the states clustered in this way relate to reaction times and trial outcomes. One can see that the most salient features demonstrated in the main figures are reproduced here: States cluster onto specific ranges of reaction times from slow to fast, and they spontaneously map onto trial outcomes, with hit and wrong trials being more interchangeable than miss trials with either hit or wrong trials. This analysis has now been added as the second part of the new Fig. S11. This is a non-trivial result because these states and the ones from the main text do not need to be inferred from the same local minima in the loss landscape.

(5) ll. 672-676R (“mapped robustly onto behavioral trial outcomes”): This statement implies that under changes in parameters of the underlying statistical model or the data, the mapping would stay similar/the same. I think that such an analysis was not conducted. Given my comments about state selection, this statement should either be down-tuned (e.g., drop mention of “robust”), or a robustness analysis should be conducted, e.g. by running the experimental protocol on subselections of the data, followed by model fitting, state selection by the final metric (see 1,2) and summarizing the results. This might also be an addition to the synthetic validation experiment.

In line with the reviewer’s suggestion, we have eliminated the mention of robustness from this statement.

(6) L. 432L, with ref to Fig S10: This statement could be more quantitative, what is the measure of similarity here?

We thank the reviewer for this point, we now address these claims more quantitatively with the new hierarchical clustering analysis shown in the new Figs. S11 and S12 (see also our answers to Points 2 and 4 above).

(7) It would strengthen the claims if the statements in ll. 485L-575L, ll. 563R-575R were backed up by statistical tests, especially when explicit statements about significance (l. 526L) are made. In general, the paper would benefit from more statistical analysis of the reported results.

We agree with the reviewer that quantitative comparisons would be ideal here. To compare the facial features per trial outcome across species, we performed a bootstrap analysis computing the overlap of facial feature weights per trial outcome in mice versus monkeys (Lines 468-473 of the current manuscript). However, we did not include the corresponding figure because we did not believe it was informative beyond the already included results (i.e., the corresponding percentiles). As we discuss there and as one can see in the figure below, facial features for hit and miss states are significantly more similar across species than would be expected by chance, whereas the facial features for wrong states are not. This again reinforces the idea that specifically the 'wrong' state is qualitatively different in monkeys and mice.

To gain better statistical insights into the similarity between facial features predictive of hit, wrong and miss trials, respectively, instead of linear correlations we focused on separability on a single-trial basis.

Specifically, in the single-trial outcome predictions shown in Fig. 4D, we tested the distribution of the model’s incorrect guesses. When identifying which states were confused for another state by the model, we see that wrong and hit trials are misclassified more regularly than both of these trial types with miss trials. As before, for mice, wrong trials appear to be confused roughly equally for hit and miss trials, while in monkeys they are almost exclusively confused for hit trials, again suggesting that in monkeys hit and wrong states are more similar to each other than in mice. This is now added in the new Fig. S7.

(8) ll. 635-637L, where is the “minimal shaped by task history” investigated?

We address this point in Fig. S17 (previously S15), where we show that only a few facial features are significantly correlated (shown in colors) with different task history variables.

(9) ll 787-799R reference “studies of internal states ...”, “... compared to previous research ...”, but no references to literature are given.

Thank you, we have now added references to two relevant studies (Cohen & Maunsell, J Neurosci 2011 and Aswood et al, Nat Neurosci 2022).

(10) There are multiple references of overfitting or avoiding overfitting (e.g. in S10 caption), but actually no analysis that investigates this statement (e.g. in terms of a line plot where the parameter of interest changes along the x-axis, and the selection metric is plotted along the y-axis for evaluation on the train and validation set). If the claims about overfitting are correct, in S10 we should see that the train performance continues to improve, but the validation performance starts degrading once the # states become too high. Is this the case?

We agree with the reviewer that our claims of overfitting were not sufficiently supported in the previous version of the manuscript. To that end, we have reframed these bits (L268-270, caption of Fig. S10) in terms of model generalization; particularly, we see that our mice model generalizes across animals. As one can see, model performance on held-out individuals was consistently high - in fact, it was indistinguishable from overall model performance for all mice, as reported in the main text. In addition, states were distributed across different ranges of reaction times in highly similar ways across sessions as well as animals, further confirming that the states we identified generalized well across animals, both in predictive performance and in their behavioural features.

These findings, now added as Fig. S22, coupled with the ones related to state fragmentation (Figs. S10, S11 and S12), provide enough evidence in favor of keeping the number of states as 4 and 3 (for macaques and mice, respectively).

- l. 23, foundational, l. 29 fundamental -> would pick one, fundamental seems more appropriate?

Thank you, we have changed 'foundational' to 'fundamental'.

- Fig S14, typo on y label: "R" instead of "R2" as in the caption.

Thank you, we have corrected this.

- What is the x-axis in S17, the norm of the feature weight?

The x axis shows the magnitude of the feature weight (i.e., its signed norm). We have clarified this (now Fig. S19).

- Caption S4, "converges to the ground truth value" -> this does not seem correct, it converges to a slightly lower value in S4B. Maybe "approaches" is a better choice of words, "converges" implies that it actually converges towards the value in the limit of many EM steps, that does not seem to be the case.

Thank you, we have corrected this in the refactored Fig S4.

- Colormap of Fig S4D could match the colormap of the state mapping in Fig 2A.

Thank you for pointing this out, we have updated the color scheme accordingly.

- Is the red/green colormap used in the figure colorblind friendly? An additional consideration for accessibility might be to also add numbers to the plot in Fig 2A (and related plots).

Thank you for pointing this out, we have tested this figure for color blindness using this freely available tool: https://bioapps.byu.edu/colorblind_image_tester and it predicts that "this image is friendly to someone with 80% deuteranopia, with 99.98% confidence"

> We already depict the input device for the monkey in Fig. 1B (top row, spherical trackball), maybe the reviewer mistook this for a movie instead of a monkey hand and trackball schematic?

> You are right and I can actually see this in the digital PDF version, but strangely, in the printed version of the manuscript I used for reviewing, this (and only this) element is missing (also in the revised version; might be an issue with my printer, but potentially good to double check).

Thanks for letting us know, we will alert copy editors of this potential issue before publication.

> We agree with the reviewer that it was difficult to precisely compare the plots visually, and have now added rounded numbers into the cells of the matrices as suggested.

> Thanks for this addition. Could you add these numbers also in Fig. S16? The number of digits might still be unified (e.g. in X.XX% format, or by keeping the # of significant digits or total digits the same?)

As the reviewer suggested, we have added the numbers, all rounded to 2 significant digits (now Fig. S18).

> We agree with the reviewer that this statement needed to be demonstrated more directly. To this end we have added the new Supp. Figs. S5 and S13. ... This again indicates that information is distributed across multiple facial features.

> I really like this addition. When looking at Fig. S5, I agree with the interpretation. Figure S13 is also a good control for the pupil size result. Minor suggestion: I would consider adding an early reference to Fig. S13 into the text in lines 1318-1327 to address this directly?

Thank you for this suggestion, we have now added a reference to this figure in L1245-1247.

Regarding the A-statistic, is it possible to add the corresponding statistical test (visually it looks like only the pupil size for A vs. B might be significant)?

We thank the reviewer for this suggestion, but we aren't sure about whether it would provide much clarity to the reader. The reason is that each of the points conforming the "face separability distribution" is, on its own, derived from a statistical test when comparing two distributions (for a given face feature over two states). Thus, we believe it mostly makes sense to compare these separability scores with respect to common reference values (shaded regions), rather than computing its p-value.

> We are very sorry, but we would need a more detailed description of this suggestion to put it into action, as we are not sure which histograms and state distributions are being referred to here.

> Apologies for being unclear. This has been already addressed with the addition made to Figure 3. I was interested in seeing an overview of how many trials are associated with each state. The panel "#trials/session vs. State" in Fig 3B is sufficient for that.

Great, we are happy this issue is resolved!

Remarks on code availability:

- The python version used is not specified. I tried to install the code with python 3.8,3.9,3.10,3.11 and 3.12, but ran into dependency issues in all cases; one common problem was the installation of the "dynamax" package. If exact dependencies are required, a conda env or docker container could clarify the exact build setup. Otherwise, clarifying the python version plus updating the requirements to successfully solve all dependencies would be helpful. Apologies if I missed something obvious.
- The code is otherwise mostly clear and commented, and the structure is clear.
- The README needs to be revised. The order of operations (e.g. installing the dependencies before cloning the repository) is not ideal. Also, some of the code files have been moved into folders, but this is not reflected in the example commands provided.
- Jupyter notebooks are added to the repository, but I think not updated according to the revision and new figures. It would be great if the authors could ensure that the code is in sync with the current version of the manuscript.
- It would be a great addition if the Jupyter notebooks were better commented. Minimally, labels to the figures could be added (optionally, along with the caption text), it is otherwise hard to navigate the document. Another suggestion would be to split the plotting notebook by figure (optional!).

Thank you for these very helpful points, we have updated the code, the README and the requirements file according to the reviewer's suggestions.

Response to Reviews:

Inferring internal states across mice and monkeys using facial features

Alejandro Tlaie, Muad Y. Abd El Hay, Berkutay Mert, Robert Taylor, Pierre-Antoine Ferracci, Katharine Shapcott, Mina Glukhova, Jonathan W Pillow, Martha Havenith, Marieke Schölvinck

Reviewer #1 (Remarks to the Author):

Thank you very much for performing the leave-one-out analysis, it is clear the model performs well. Perhaps it could be added to Figure 2A as an additional whisker on the plot, it is a bit unfortunate to leave an important analysis for Figure S22. And thank you for the additional pupil analysis, it is an interesting finding that is well-supported.

We thank the reviewer for this encouraging feedback. We tried adding a plot to Figure 2A, but could not find a way for the results to be clearly readable and clearly distinct from the R2 analyses already shown in the current figure. Nevertheless, we agree that this is an important result. To make these results more prominently accessible, we have merged Supp. Fig. S22 with the previous Supp. Fig. S8, as both figures address how the model generalises across animals. We have also added a passage in the main text highlighting these analyses more clearly (Lines 267-275). We hope these changes improve accessibility to this analysis sufficiently.

Reviewer #2 (Remarks to the Author):

I thank the authors for the replies to my comments and the additional analysis. I especially appreciate the new figures and analysis aimed to address the comments about the synthetic data setup.

In summary, my biggest concern in the previous review was the mismatch in the validation of the main analysis method on synthetic vs. real data. This synthetic validation used a different model selection protocol (for the number of states) than what the authors used on the real dataset.

The authors now revised Figure S4A to use a validation protocol in which internal state number is by a hierarchical clustering method; on the synthetic data, selecting the best performing models according to cross-validated R2 then over clusters the number of states, selecting 10 instead of 5. This is followed by a hierarchical clustering step to group states together, and uses a shuffle control to determine the number of appropriate states. This selection mechanism is able to discover the correct number of ground truth states with acceptable correlation to the ground truth states.

The authors then re-analysed (in the supplement, not the main paper) their dataset using this selection protocol and show support for 2-4 states (monkey) or 2-4 states (mice), and the states reported in the paper would fall into these ranges. It remains debatable if the resulting states in Fig. S12 (state 1, Wrong) and Fig. 4A (state A, Wrong) are fully comparable/well recovered: The former has a median reaction time around $\sim .2s$, the latter around $\sim .5s$ (eyeballed from the plots). This is partially covered by statements in the main paper, which make less strong claims about this “Wrong” state.

In any case, the additions to the data analysis in my opinion sufficiently support findings resulting from the state selection, and where deviating interpretations are possible. I thank the authors for the additional time invested in better contextualising their analysis results.

We thank the reviewer for their very helpful suggestions and encouraging comments. We believe the analyses added based on their comments have substantially improved the manuscript and are grateful for their thoughtful feedback.

Further minor comments, mostly related to these additions:

- l. 1212 “Figure S4A illustrates the input data” -> seems to be a typo due to reorganisation of plots, there is no data referenced in that figure
- l. 1214 “does peak at the ground truth one” -> “does converge to the ground truth one” ?
- l. 1215 “for some example trials” -> why not for all?
- l. 1219 “there is an almost perfect match ... > .9” -> the figure shows ranges from .73 to .91, this statement seems to be inaccurate.
- Fig S4D: What about the off-diagonal elements? Right now, it looks like all these correlations would be 0, but that does not appear plausible.
- Fig S4, caption: “approaches to” / “approaches”
- l. 1245-1247 these statements reference “significant” or “not significant” results, but do not state a p-value and the used test. The referenced figure also contains no details about this.

We thank the reviewer for these pertinent remarks. We have addressed all of them (see Lines 1212-1228) by including a more detailed explanation of the hierarchical clustering procedure in a separate sub-section (Lines 1229-1231).

Reviewer #2 (Remarks on code availability):

I thank the authors for revising and improving the code. I did not attempt to re-run the code, but noticed a few minor remaining issues, e.g. in the README the path to the requirements file is not exactly matching, or there are duplicate files in the repo, like "mouse_MSLR_final copy.py" vs. "mouse_MSLR_final.py". A final cleanup and check would be good.

We thank the reviewer for these pertinent remarks and for taking the time to diligently check all of the crucial steps within the pipeline. We have updated the repository and made sure that there are no file duplicates, as well as checking the folder structure instructions within the README.